# Temperature measurement of Quark-Gluon plasma at different stages

STAR Collaboration* ✉

In a Quark-Gluon Plasma (QGP), the fundamental building blocks of matter, quarks and gluons, are under extreme conditions of temperature and density. A QGP could exist in the early stages of the Universe, and in various objects and events in the cosmos. The thermodynamic and hydrodynamic properties of the QGP are described by Quantum Chromodynamics (QCD) and can be studied in heavy-ion collisions. Despite being a key thermodynamic parameter, the QGP temperature is still poorly known. Thermal lepton pairs ($e^+e^-$ and $\mu^+\mu^-$) are ideal penetrating probes of the true temperature of the emitting source, since their invariant-mass spectra suffer neither from strong final-state interactions nor from blue-shift effects due to rapid expansion. Here we measure the QGP temperature using thermal $e^+e^-$ production at the Relativistic Heavy Ion Collider (RHIC). The average temperature from the low-mass region (in-medium $\rho^0$ vector-meson dominant) is $(2.01 \pm 0.23) \times 10^{12}$ K, consistent with the chemical freeze-out temperature from statistical models and the phase transition temperature from Lattice QCD. The average temperature from the intermediate mass region (above the $\rho^0$ mass, QGP dominant) is significantly higher at $(3.25 \pm 0.60) \times 10^{12}$ K. This work provides essential experimental thermodynamic measurements to map out the QCD phase diagram and understand the properties of matter under extreme conditions.

The state of QCD matter is typically characterized by its temperature and baryon chemical potential[1,2], as depicted in Fig. 1. In standard thermodynamics, the baryon chemical potential, $\mu_B$, is a measure of the change in free energy due to an increase of baryon number by one in a fixed volume, and increases monotonically with net baryon density. Dielectrons, i.e., electron-positron pairs, are excellent thermometers of the extremely hot and dense QCD matter[3–6] created in high-energy heavy-ion collisions. Leptons do not participate directly in the strong interactions; consequently electrons and positrons have minimal interactions with the predominantly strongly-interacting particles throughout the evolution of the system in both its initial quark-gluon and its final hadronic states[7–11]. As is the case for any black-body radiation spectrum, higher temperatures yield harder dielectron energy and mass spectra, i.e., exhibit an increase in the ratio of high to low mass pairs[7,8]. QGP consists of quarks and gluons with temperatures in excess of hundreds of MeV(where 100 MeV corresponds to

$1.16 \times 10^{12}$ K), higher than the QCD critical temperature ($T_C$)[12–14]. Theoretical studies[15,16] from lattice QCD (LQCD) predict a crossover transition with a smooth but rapid change of thermodynamic quantities in a narrow region around $T_C \sim 156.5$ MeV at $\mu_B < 300$ MeV. Following the initial stage of a heavy-ion collision, the system cools down as it expands rapidly. Throughout its expansion, the hot system radiates both photons and lepton pairs which can be measured by dedicated particle detectors. As a result, different ranges of the dielectron energy and mass spectra are dominated by the radiation at the different stages of the system's evolution. Photon momentum spectra have been used to determine the QGP temperature at the Relativistic Heavy Ion Collider (RHIC) and the Large Hadron Collider (LHC) during the past two decades[17–20]. However, an unambiguous interpretation is complicated by the rapid bulk expansion at those collision energies because the expansion velocity is a substantial fraction of the speed of light and it alters the energy spectrum of the photons. This blue-shift effect makes

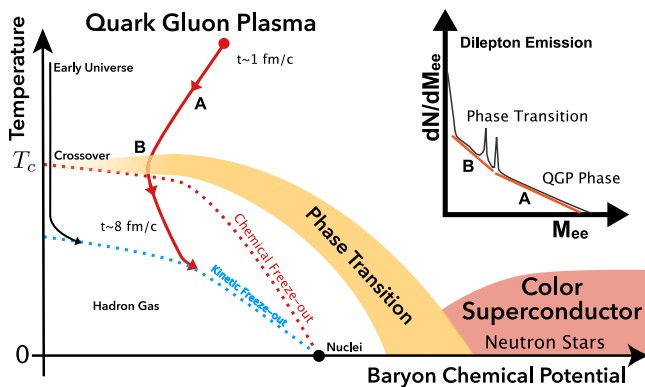

**Fig. 1 | A schematic view of $e^+e^-$ pair production and the QCD phase diagram.** The diagram illustrates matter properties with baryon chemical potential (equivalent to net baryon number density) and temperature, with landmarks of normal nuclei, neutron stars, and the phase transition to QGP. A heavy-ion collision creates a QGP at high baryon chemical potential and high temperature shortly after the initial impact, and the system evolves along the red arrow toward the phase boundary and hadronization. The insert depicts the dilepton spectrum with low-mass and intermediate-mass ranges corresponding to the dominant emission contribution from the transition and the QGP phases, respectively.

it very difficult, if not impossible, to extract the true "blackbody" radiation temperature from the detected photon energy spectrum and overly reliant on model assumptions[11]. On the other hand, electron-positron pairs from thermal emissions provide an additional degree of freedom through the reconstruction of their invariant mass, $M_{ee}$, a frame-independent variable[7–10]. The invariant-mass spectrum of thermal dielectrons is immune to blue-shift effects and is thus able to provide a true measurement of the temperature of the QGP at different stages of the evolution. In the early stage of QGP evolution, thermal dielectrons are predominantly produced via the annihilation processes among quarks and anti-quarks, and potentially gluons at higher order[21,22]. During the phase transition, as the system cools down, deconfined quarks begin to hadronize into colorless baryons and mesons. The resulting strongly-interacting mixed medium, with both partonic and hadronic degrees of freedom, still exhibits bulk thermodynamic and hydrodynamic properties and continues to expand and cool down. At this stage and later on, dielectrons primarily arise from the decay of $\rho^0$(770) vector mesons produced inside the medium. The dense hadronic medium continuously creates $\rho^0$ mesons through frequent hadronic interactions. The $\rho^0$ meson with a lifetime of about 1.3 fm/$c$ mostly decays inside the medium that lasts for tens of fm/$c$[11]. Consequently, the invariant-mass spectrum of the in-medium $\rho^0$ reconstructed via dielectrons is considered an excellent experimental probe of the dissolving hadronic mass lineshape close to the QCD phase transition[23–26]. In the subsequent evolution, the system experiences a stage in which the inelastic interactions among particles cease due to decreasing density, resulting in the freezing of particle composition. This stage is known as the chemical freeze-out[27]. Elastic interactions continue after freeze-out, and these influence particle momenta. At the very last stage of the expansion, elastic interactions stop, and the system enters the stage of kinetic freeze-out[28].

During the past three decades, measurement of the thermal dilepton production in heavy-ion collisions has been an essential scientific program of several experiments conducted at particle accelerator facilities such as the Bevalac[29], the SIS18[30], the SPS[31–35], RHIC[36–40], and the LHC[41]. These experiments cover a wide range of nucleon-nucleon center-of-mass energies, $\sqrt{s_{NN}}$, spanning from 2.20 GeV to 5.02 TeV. The measured dilepton spectra in the low-mass region from SPS to RHIC energies have provided strong experimental evidence that the spectral function of the in-medium $\rho^0$ vector meson is substantially broadened without significant change of its mass peak[23–26]. The HADES

experiment has recently shown that the thermal dielectron mass spectrum exhibits a near-exponential fall-off in Au+Au collisions at $\sqrt{s_{NN}}$ = 2.42 GeV[30]. Although its kinematic reach is below the $\rho^0$ pole mass (775 MeV/$c^2$), the result indicates that the $\rho^0$ resonance spectrum is significantly altered by the frequent interactions among the baryons in the dense hadronic medium and the process could produce a seemingly thermalized system. The average temperature extracted from this exponential spectrum was determined to be 71.8 ± 2.1 MeV. In the higher mass region (above 1 GeV/$c^2$), the NA60 experiment reported a benchmark result of thermal dimuon production in In+In collisions at $\sqrt{s_{NN}}$ =17.3 GeV[33]. The temperature extracted from the thermal dimuon spectra is found to be 200 ± 12 MeV[42], which is significantly higher than the $T_C$, providing the first direct evidence that these thermal dileptons are emitted from the deconfined partonic medium. Heavy-ion collisions at various energies allow for the exploration of the QCD phase diagram with diverse trajectories in terms of temperature and $\mu_B$. Nevertheless, the dileptons originating from decays of open charm ($c$ and $\bar{c}$ quarks) exhibit substantially higher yields than thermal dileptons radiated from the QGP, resulting in a lack of reliable QGP temperature measurements for both RHIC and LHC top energies. During the RHIC Beam Energy Scan Phase I (BES-I), STAR conducted measurements of thermal dielectrons at lower energies[37,39], where the influence of open charm decays is significantly reduced. Despite this advantage, the statistical precision limitations of these measurements have hindered the extraction of QGP temperature values. There exists one intermediate-mass region ($m_\phi \lesssim M_{\mu\mu} \lesssim m_{J/\psi}$) measurement from NA60[33] at one beam energy to date. In this article, we present dielectron spectra in Au+Au collisions at $\sqrt{s_{NN}}$ = 27 and 54.4 GeV using large datasets from the STAR detector at RHIC, collected in years 2017 and 2018. The temperatures of hot nuclear matter in the low-mass and intermediate-mass regions are extracted from the thermal dielectron distributions. These results provide unique access to the thermodynamic properties at both the early stage of the QGP phase, and the late stage near the phase transition to hadronic matter via different trajectories within the QCD phase diagram.

## Results and Discussions

The most relevant subdetectors of the STAR detector are depicted in Fig. 2 (top panel) together with a typical event display from a heavy-ion collision. Charged particles produced in these collisions leave ionization trails inside the Time Projection Chamber (TPC)[43]. The radius of curvature of a charged particle trajectory in an externally applied magnetic field (B = 0.5 Tesla) is used to determine its momentum per charge ($p/q$). The ionization energy loss per unit length ($dE/dx$) along a particle's path through the TPC gas as a function of its $p/q$ is shown in Fig. 2. Combining the time-of-flight information measured by TOF with the path length and momentum information, the mass ($m$) of charged particles can be obtained. The electrons of interest have the typical characteristics of a relativistic rise in $dE/dx$ and low $m^2$ as shown in Fig. 2. These two powerful particle identification techniques provide high-purity electron identification with a hadronic background rejection rate of more than 5 orders of magnitude and a large fiducial acceptance[38,44]. The identified electrons and positrons from the same event are combined to reconstruct the invariant mass and transverse momentum of all possible pairs. However, more than 99% of these pairs are random combinations, commonly referred to as the combinatorial background, which need to be subtracted in order to obtain the inclusive dielectron signals[38]. Following corrections for the pair reconstruction efficiency and acceptance, the fully corrected inclusive dielectron signal is established as shown in the top two panels of Fig. 3. More details on these and other analysis procedures can be found in the Methods Section.

The measured inclusive dielectron spectra are an accumulation of contributions from various stages throughout the evolution of the system following a heavy-ion collision. These include dielectrons

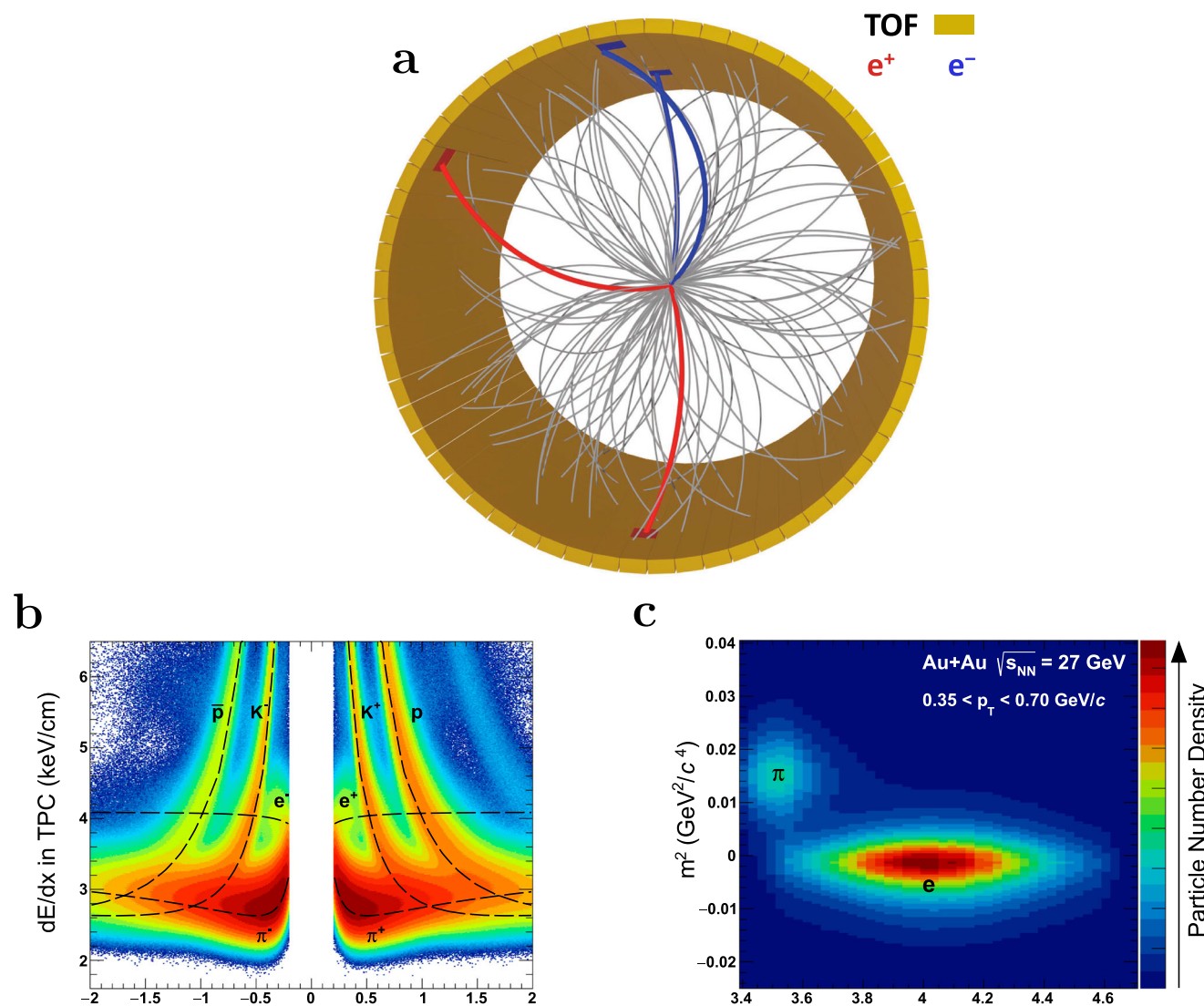

**Fig. 2 | A schematic display of a Au+Au collision reconstructed with the STAR detector. a** Charged-particle ionization in the gas of the TPC, forming three-dimensional tracks (gray lines) that curve due to the magnetic field. As tracks exit the outer radius of the TPC, they leave signals (red and blue hits) in the TOF detector. Electrons (blue) and positrons (red) tracks are identified based on the ionization energy loss $dE/dx$ and mass squared $m^2$ measured by the TPC and TOF. **b** $dE/dx$ as a function of momentum per charge $p/q$. **c** $m^2$ vs. $dE/dx$ distribution of the electron candidates in a transverse momentum interval of $0.35 < p_T < 0.70$ GeV/$c$. The dashed curves represent the $dE/dx$ values expected from Bethe-Bloch (Bichsel) equation[92] for the respective particles.

from the thermal QCD medium of the collision and also from non-thermal physical sources. At the very early stages, dielectrons are produced through the Drell-Yan processes[45] in which quarks and anti-quarks from the colliding nucleons annihilate through virtual photons into lepton pairs. At much later stages, after the hot medium has disintegrated, dielectrons are produced from the decays of (relatively) long-lived hadrons. These include the two-body decays from $\omega$, $\phi$, J/$\psi$ $\to e^+e^-$, Dalitz decays[46] from $\pi^0$, $\eta$, $\eta'$, J/$\psi$ $\to \gamma e^+e^-$ and $\omega \to \pi^0 e^+e^-$, $\phi \to \eta e^+e^-$, and the weak, semi-leptonic decays of open-charm hadrons. The contributions from these physics backgrounds are commonly referred to as the "cocktail" and can be well determined from simulations, shown in the top two panels of Fig. 3. The fully corrected data substantially exceed the total physical background "Cocktail Sum" over a large mass region due to significant contributions from thermal dielectrons at lower masses. To quantify the thermal component, the excess dielectron mass spectrum is obtained by subtracting the cocktail sum from the measured, fully

corrected inclusive data. Further details about the cocktail simulations can be found in the Methods Section.

The measured invariant-mass spectra of the thermal dielectrons (i.e., the excess dielectrons) are shown in the bottom panel of Fig. 3. The spectra are normalized by the charged particle multiplicity at mid-rapidity $dN_{ch}/dy|_{y=0}$ in order to compare the measurements among different colliding species and beam energies. Two important invariant-mass ranges in this study are defined as follows: the low-mass region (LMR), $0.4 < M_{ee} < 1.20$ GeV/$c^2$, and the intermediate-mass region (IMR), $1.0 < M_{ee} < 2.9$ GeV/$c^2$. The bottom panel of Fig. 3 shows the LMR results where the STAR Au + Au collision data at $\sqrt{s_{NN}} = 27$ and 54.4 GeV are consistent with each other within the entire mass region. The STAR LMR data also show good agreement with the dimuon results from In+In collisions at $\sqrt{s_{NN}} = 17.3$ GeV, while the Au+Au IMR central values are systematically higher than the NA60 data. These observations suggest that in the LMR, the thermal dileptons from the three measurements originate from radiative sources with a similar

a

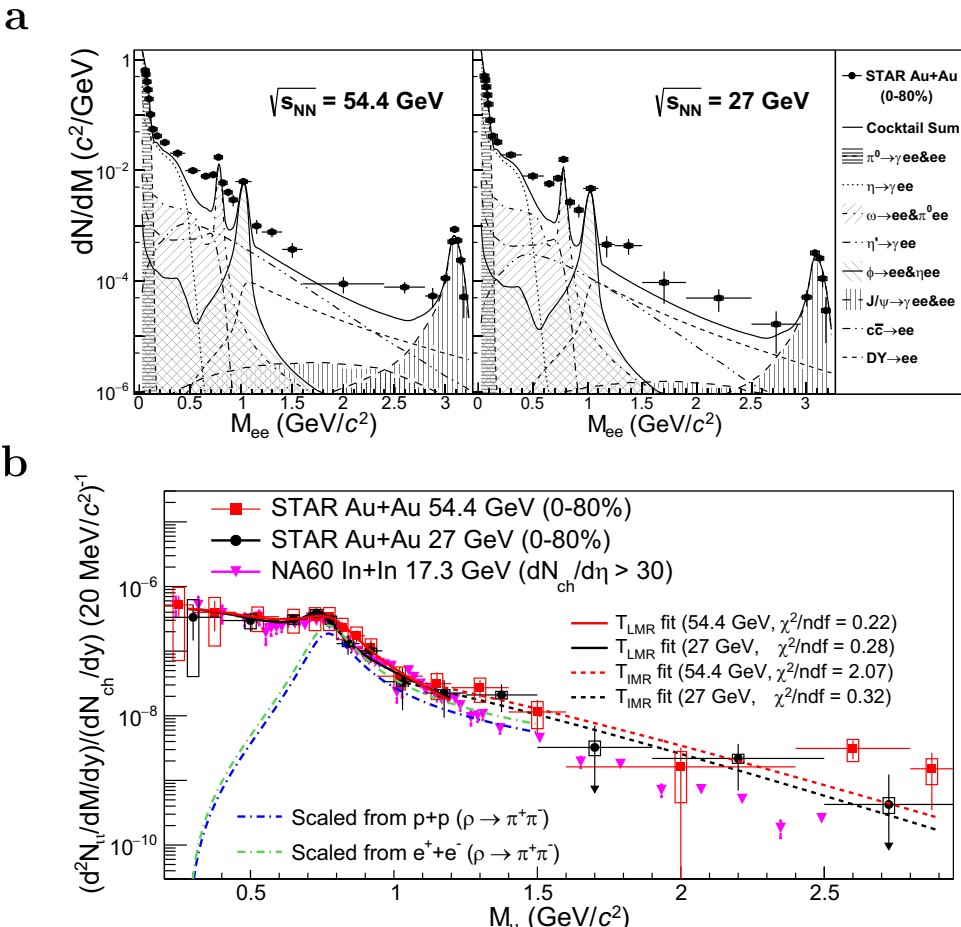

b

**Fig. 3 | Dielectron invariant-mass spectra. a** Fully corrected inclusive dielectron mass spectra (black dots) compared to the physics background (dashed lines and shaded lines) in Au+Au minimum-bias (0-80% centrality) collisions at $\sqrt{s_{NN}}$ = 54.4 and 27 GeV. **b** Thermal dielectron mass spectra from 54.4 GeV (red squares) and 27 GeV (black dots) compared to the NA60 thermal dimuon data (magenta inverted triangles). *ll* denotes the dielectron or dimuon pairs. Dashed lines show the fitting curves for the corresponding temperature extractions. Dot-dashed lines display the expected vacuum $\rho$ spectra ($f^{BW}(M)$) based on the p+p[48] and $e^+e^-$[49] collision data. Vertical bars and boxes around data points represent the statistical and systematic uncertainties, respectively. Downward arrows indicate statistical uncertainties exceeding 100%.

temperature, while the thermal dileptons in the IMR may indicate sources with different temperatures in these three different energies. To quantify the temperature of the thermal source responsible for LMR radiation, a function that combines the in-medium resonance structure and the continuum thermal distribution is used to fit the measured mass spectrum. The mass lineshape of $\rho^0$ decaying to dileptons in vacuum can be described by a relativistic Breit-Wigner function[47–49], $f^{BW}(M) = \frac{MM_0\Gamma_{ll}}{(M_0^2 - M^2)^2 + M_0^2\Gamma^2}$, where $M$ is the invariant mass of the dilepton pair, $\Gamma = \Gamma_0 \frac{M_0}{M} \left(\frac{M^2 - 4m_\pi^2}{M_0^2 - 4m_\pi^2}\right)^{3/2}$ is the total width, predominantly influenced by the $p$-wave decay $\rho^0 \to \pi^+\pi^-$, and the $\rho^0 \to l^+l^-$ decay width[50,51] $\Gamma_{ll} \propto \left(1 + \frac{2m_l^2}{M^2}\right)(1 - 4m_l^2/M^2)^{1/2}$. Here, $M_0$ and $\Gamma_0$ are the pole mass and width of $\rho^0$ meson, while $m_l$ is the lepton mass. The dilepton yields from these in-medium $\rho^0$ meson decays are proportional to $f^{BW}(M)$ multiplied by $M^{3/2} e^{-M/k_B T}$[34,35,52,53]. Both $f^{BW}(M)$ and the Boltzmann factor ($e^{-M/k_B T}$) are highly dependent on the medium temperature. If the $\rho^0$ is completely dissolved in the medium, its mass spectral structure ($f^{BW}(M)$) spreads out and approaches a smooth distribution similar to a $q\bar{q}$ continuum (QGP thermal radiation)[8,30] which can be directly described by $M^{3/2} e^{-M/k_B T}$. The extracted temperatures $T_{LMR}$ from the LMR thermal dielectron mass spectra are 165 ± 20(stat.) ± 21(syst.) MeV and 178 ± 15 (stat.) ± 13 (syst.) MeV for

the Au+Au collisions at $\sqrt{s_{NN}}$ = 27 and 54.4 GeV, respectively. A similar fit to the NA60 data (shown in the Methods Section) gives a temperature of 172 ± 6 MeV. The temperatures extracted from the LMR thermal dileptons of different collision energies and species are consistent with each other and in agreement with the conjecture (discussed in the previous section) that they are radiated from thermal sources with a similar temperature. The IMR results for the 27 GeV and 54.4 GeV Au+Au collisions are also consistent within their uncertainties. The mass spectrum in this mass region is smooth, and the temperature is extracted by fitting the Boltzmann function $M^{3/2} e^{-M/k_B T}$[8]. The extracted temperatures $T_{IMR}$ for the 27 GeV and 54.4 GeV Au+Au collisions are 274 ± 65 (stat.) ± 10 (syst.) MeV and 287 ± 70 (stat.) ± 34 (syst.) MeV, respectively. By fitting the thermal dimuon spectra of In+In data in this mass region (shown in the Methods Section), a temperature of 245 ± 17 MeV is extracted. For all the systems, the extracted temperature for the low-mass range is significantly lower than that of the intermediate-mass range. This observation is consistent with the expectation that LMR thermal dileptons are predominantly emitted at a later stage of the medium evolution around the phase transition, while those in the IMR are influenced by the earlier partonic stage with higher temperatures. Although theoretical studies[8,10,23,24] anticipate a hotter QGP created at higher collision energies, this trend is not apparent in our data due to the current precision limitations. It should

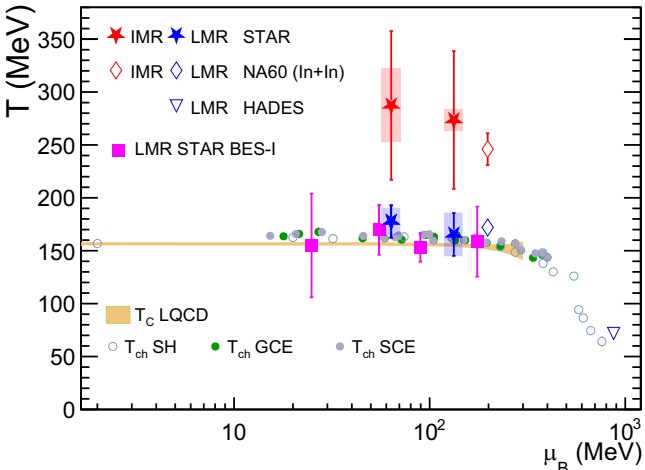

**Fig. 4 | Temperatures vs. baryon chemical potential.** Temperatures extracted from in-medium $\rho^0$; the region of the later QGP stage (blue stars), and the region of the earlier QGP stage (red stars) from STAR data are compared to the temperatures extracted from NA60 data[33] (diamonds) and HADES data[30] (inverted triangle). Chemical freeze-out temperatures extracted from the statistical thermal models (SH, GCE, SCE)[27,55] are shown as open and filled circles. The QCD critical temperature $T_C$ at finite $\mu_B$ predicted by LQCD calculations[15] is shown as a yellow band. All temperatures are plotted at the $\mu_B$ determined at chemical freeze-out. Vertical bars and boxes around the data points represent the statistical and systematic uncertainties, respectively.

be noted that these temperature values from the IMR are systematically higher than the TAMU model estimations using entropy and adiabatic thermodynamic expansions[8]. Radiation at an early stage, when the system is in a non-equilibrium state, may contribute to the IMR dielectrons and yield a higher apparent temperature[54]. Future experiments with enhanced statistical and systematic precision, alongside further theoretical studies, are necessary to clarify these observations.

Figure 4 summarizes the temperature measurements as a function of the baryon chemical potential $\mu_B$. The chemical freeze-out temperature $T_{ch}$ and $\mu_B$ can be well determined by applying statistical thermal models[27] to the hadron production yields. The chemical freeze-out temperatures extracted from various statistical thermal models (SH, GCE, SCE)[27,55] are shown in the figure as filled and open circles. Similarly, temperatures were extracted from previously published low-mass thermal dielectron spectra[37,39] with a thermal distribution of $M^{3/2}e^{-M/k_BT}$. The extracted $T_{LMR}$ from the limited statistics of those measurements are in good agreement with STAR's new results from Au+Au collisions at $\sqrt{s_{NN}}$ = 27 and 54.4 GeV as well as the result extracted from NA60 In+In data at $\sqrt{s_{NN}}$ = 17.3 GeV. Moreover, all these $T_{LMR}$ values are found to be consistent with $T_C$ and $T_{ch}$.

A long-standing challenge has been the empirical observation that the $T_{ch}$ extracted from the yields of the final-state hadrons coincides with the QCD phase transition temperature ($T_C$) from LQCD[27]. Stable hadrons emerge from the chemical freeze-out and their yields are an integration over the whole volume of the system. Therefore, the extracted $T_{ch}$ by definition should have been lower than the phase transition temperature. The present dilepton measurements can provide new insight towards resolving this puzzle. The measured yields of the low-mass thermal dileptons are an accumulation from the initial QGP stage to the final kinetic freeze-out. Therefore, these yields are integrated over the whole system volume and over the entire radiative evolution time. These measured dilepton yields can be compared to those from $\rho^0$ decays in the vacuum at chemical freeze-out estimated from the two baseline measurements of the $\rho^0$ yields from its $\pi^+\pi^-$ decay channel in proton-proton and $e^+e^-$ collisions. A comparison of the charged-particle

multiplicity ($dN_{ch}/dy|_{y=0}$) normalized dilepton yields measured in heavy-ion collisions with the expected dilepton yields at freeze-out clearly shows that the measured dielectron yields (within 0.4 – 0.75 GeV/$c^2$ mass window) are more than a factor of 5 larger than those two baseline yields (cf. Fig. 3 bottom panel and Methods Section). The high yields of dileptons, the strong in-medium broadening of the $\rho$ spectral function, and the approximate overlap of $T_C$, $T_{ch}$ and $T_{LMR}$ at these energies suggest that the low-mass thermal dileptons are predominantly emitted over a long period of time at high density around a fixed temperature. Such a scenario is possible under the assumption of strong influence by a phase transition[8,56] and/or by a soft point in the equation of state[57,58]. These measurements provide a direct experimental tool for accessing the temperature in the vicinity where the phase transition to deconfinement occurs - one of the most fundamental landmarks of the QCD phase diagram.

Thermal dileptons were proposed to serve as a critical thermometer of QGP created in high-energy heavy-ion collisions. However, their production rate is very low and the physics background is large. The extraction of temperature from the dilepton spectra has been limited to a single beam energy in a relatively small collision system performed by the NA60 experiment. By studying Au+Au collisions at various energies, we explore the QCD phase diagram across different temperatures and baryon chemical potentials. For the first time, we report the average QGP temperatures at two stages of their time evolution and at multiple baryonic chemical potentials, advancing our understanding of QGP thermodynamic.

## Methods
### Data description
The thermal dielectron measurements are based on the datasets collected with the STAR detector in Au+Au collisions at $\sqrt{s_{NN}}$ = 27 GeV (year 2018) and 54.4 GeV (year 2017), using the minimum-bias (MB) trigger which requires a coincidence of signals in the opposite beam-going direction ($-z$ and $+z$ components of either the Vertex Position Detector[59] (VPD, 4.25 < $|\eta|$ < 5.1), Beam-Beam Counters[60] (BBC, 2.2 < $|\eta|$ < 5.0) or the Zero Degree Calorimeters[61] (ZDC, $|\eta|$ > 6.0). To ensure the quality of event reconstruction, requirements on the primary vertex reconstructed via the Time Projection Chamber[43] (TPC) detector along the beam axis ($|V_z^{TPC}|$ < 35 cm) and the transverse radial axis ($V_r^{TPC}$ < 2 cm) are applied. For the pile-up event rejection, correlations between the number of hits in the Time Of Flight[62,63] (TOF) and the reference multiplicity of TPC tracks are considered for $\sqrt{s_{NN}}$ = 27 GeV data, while the difference between $V_z^{TPC}$ and $V_z^{VPD}$ measured by the VPD is required to be within 3 cm for $\sqrt{s_{NN}}$ = 54.4 GeV data. There are 256 M and 500 M events for $\sqrt{s_{NN}}$ = 27 GeV and 54.4 GeV Au+Au collisions, respectively, that satisfy the event selections.

### Reconstruction of e⁺e⁻ pairs
The electrons and positrons are identified via the TPC detector (tracking, momentum and $dE/dx$) and TOF detector (time), as described in refs. 37,38. These electrons and positrons are required to be within the STAR detector acceptance of pseudo-rapidity ($|\eta^e|$ < 1) and transverse momentum ($p_T^e$ > 0.2 GeV/$c$). The selected electrons and positrons from the same events are combined to reconstruct the unlike-sign dielectron ($e^+e^-$) pairs. The raw yields of these inclusive $e^+e^-$ pairs are denoted as $N_{+-}$. The background in the inclusive unlike-sign pairs arises from uncorrelated (random combinatorial) and correlated background (e.g. jet fragmentation) pairs[64]. These background contributions can be well reproduced by the geometric mean of the like-sign pairs $N_{geomLS} = 2\sqrt{N_{++} \times N_{--}}$ as demonstrated in ref. 64, where the $N_{++}$ and $N_{--}$ represent the raw yields of the like-sign pairs $e^+e^+$ and $e^-e^-$ reconstructed by electrons or positrons in the same event. The background pairs from photon conversion in the detector materials

are removed by the $\phi_V$ angle selection method developed by the PHENIX Collaboration[64]. The raw yield of inclusive dielectron signals can be calculated as:

$$N_{raw} = N_{+-} - N_{geomLS} \times f_{sign}, \tag{1}$$

where $f_{sign}$ is a correction factor accounting for the differences of detector acceptance between unlike-sign and like-sign particle pairs due to the magnetic field. This factor is evaluated by the ratio of the unlike-sign to the like-sign pairs using event mixing techniques[38,64],

$$f_{sign} = \frac{B_{+-}}{2\sqrt{B_{++}B_{--}}}, \tag{2}$$

where $B_{+-}$, $B_{++}$ and $B_{--}$ represent the number of unlike-sign pairs and like-sign pairs in the mixed events, respectively. $B$ and $N$ are measured as 2-dimensional functions of $M_{ee}$ and $p_T$. The $N_{+-}$, $N_{geomLS}$, $N_{raw}$ and signal over background ratio are displayed in the Supplementary Fig. 1. The ratio of signal to combinatorial background is about 1:100 or 1:200.

### Efficiency and acceptance corrections

In this study, the single electron reconstruction efficiency includes the tracking reconstruction efficiency (TPC) and the electron identification efficiency (TPC and TOF). The tracking efficiency is evaluated using Monte Carlo simulation embedding techniques, while the electron identification efficiency is determined using data-driven techniques as described in ref. 38. The dielectron pair reconstruction efficiency correction ($\epsilon^{pair}$) and acceptance correction (Acc$^{pair}$) are calculated through the virtual photon decay simulation. This simulation incorporates all the single-electron efficiencies for each daughter particle in a full 3D momentum space of ($p_T$, $\eta$, $\phi$). The pair reconstruction efficiency and acceptance corrections are calculated and then applied to correct the raw signal yields in 2D ($M_{ee}$ vs. $p_T$). The corrected data are normalized by the number of events used for the raw data reconstruction to obtain the invariant yields, which represent the production rate of the reconstructed signal per Au+Au collision in a given centrality class. The correction factors for the dielectron pair efficiency and acceptance are displayed in Supplementary Fig. 2.

### Physics background from non-thermal sources

The background of dielectron pairs from non-thermal physics sources (conventionally named cocktails "CKT") is determined through cocktail simulation techniques[38,64]. This process is accomplished through two major steps: (1) simulating the invariant mass lineshapes through the dielectron decay channel, and (2) scaling their contributions by their invariant yields. In the simulations, the detector acceptance and momentum resolution are incorporated into the simulations of hadron decays to accurately reproduce the background in real data. Long-lived light hadrons such as $\pi^0$, $\eta$, $\eta'$, $\omega$, and $\phi$ preserve their vacuum decay structures as they decay after the kinetic freeze-out of the collision system. The $p_T$ spectra of light hadrons are determined through the Tsallis Blast-Wave (TBW) model[65], where the model parameters are obtained from the fit to STAR's measured light hadron production[55]. The rapidity distribution of light hadrons is determined by the equation from CERES' Monte Carlo event generator, GENSIS[66], which is parameterized to match CERN SPS data[67–70]. The $J/\psi$ vector meson has a much longer lifetime ($\sim 2 \times 10^3$ fm/$c$) compared to the typical medium lifetime ($\sim 10$ fm/$c$), ensuring that almost all the $J/\psi$ decay in vacuum, regardless of their production timing. The $p_T$ spectra of the $J/\psi$ are obtained from STAR published data[71]. After the mass lineshapes of the physics background sources are determined, the next step is to scale them with the invariant yields. The $\pi^0$ yields are determined by averaging $\pi^\pm$ from high-precision STAR data[55]. The invariant yield ratios

$\sigma_\eta/\sigma_{\pi^0}$ and $\sigma_{\eta'}/\sigma_{\pi^0}$ are taken from experimental data[64,72–74]. The invariant yields of $\omega$, $\phi$, and $J/\psi$ mesons are determined in this study by taking into account the distinct differences in their mass lineshapes compared to the smooth thermal dielectron mass spectra. Since the $\omega$ and $\phi$ have a narrow peak structure compared to the $\rho$, theoretical lineshape of total thermal dielectrons from the TAMU model and the simulated $\omega$ and $\phi$ lineshapes are fit to data to extract the $\omega$ and $\phi$ yields more precisely.

The semi-leptonic decays of charmed hadrons are a special type of physics background. In heavy-ion collisions, charm and anti-charm quarks are created in pairs through initial hard interactions and then form charmed and anti-charmed hadrons with long lifetimes of $O(100)$ $\mu$m/$c$. When these hadrons decay, they produce electrons and positrons through semi-leptonic decay in vacuum. The invariant mass spectra from these electron-positron pairs are distributed smoothly and are expected to contribute significantly in the mass range of 1-3 GeV/$c^2$. The contribution from open charm decays is simulated in $p+p$ collisions using PYTHIA v6.416[75], with the settings described in ref. 76. The total cross section for charm production per nucleon-nucleon collision has been measured worldwide as a function of the center-of-mass energy $\sqrt{s}$, and the results are presented in Supplementary Fig. 3. To estimate the cross section values at $\sqrt{s} = 27$ and 54.4 GeV, the experimental data are fit with a theoretical curve from NLO pQCD calculations (MNR[77]). To estimate the uncertainties associated with the cross section values, two alternative approaches were employed to gauge the impact on the default values. Firstly, the NLO pQCD curve fit was applied exclusively to data up to RHIC energies, allowing for an assessment of the exclusion of higher energy data. Secondly, the FONLL curve, as reported in ref. 78, was directly utilized to give the extrapolations. The differences arising from these two approaches in comparison to the default values were incorporated into the total uncertainties of the cross section values. The resulting values and the associated uncertainties of $\sigma_{c\bar{c}}^{NN}$ are 16.7 ± 3.3 $\mu$b and 72.0 ± 14.4 $\mu$b for $\sqrt{s}$ = 27 GeV and 54.4 GeV, respectively. The contribution from the Drell-Yan process is simulated for the $p+p$ collisions using mainly the same PYTHIA v6.416 settings as in a previous study[76], while the $k_T$ (the intrinsic transverse momentum that partons of the colliding protons have before the hard scattering process occurs, which controls the $p_T$ distribution of final-state particles) is tuned to be 0.95 GeV/$c$ to match the measured Drell-Yan $p_T$ spectrum from the FNAL-288 experiment[79] in the mass region of 5-9 GeV/$c^2$. The total production cross section of $c\bar{c}$ and Drell-Yan process for Au+Au collisions is calculated by multiplying the average number of nucleon-nucleon binary collisions ($N_{coll}$) for a given collision centrality, as obtained from a Glauber model[80].

### Thermal e$^+$e$^-$ spectrum

The thermal dielectron spectra are determined by subtracting all the physics background from the inclusive dielectron spectra with

$$N_{thermal} = \left( \frac{1}{N_{event}} \times \frac{N_{raw}}{\epsilon^{pair}} - N_{CKTSum}^{inAcc} \right) \times \frac{1}{Acc^{pair}} \tag{3}$$
$$= N_{FullCorr} - N_{CKTSum},$$

where $N_{thermal}$, $N_{FullCorr}$ and $N_{CKTSum}$ represent the number of pairs from the thermal dielectron, the fully corrected inclusive dielectron and the total physics backgrounds. The $N_{CKTSum}^{inAcc}$ represents the amount of physics background within the STAR acceptance. The thermal dielectron production spectra are studied in various Au+Au collision centralities including the 0-80% centrality and the sub-centralities (0-10%, 10-40%, 40-80%). For each given centrality bin, the dielectron signals, the efficiency corrections, and its cocktail simulations are carried out individually for the final thermal dielectron spectrum determination.

## Systematic uncertainties

In thermal dielectron spectrum measurements and temperature measurements, the sources of systematic uncertainties arise from both measurements of data and estimation of physics background, respectively. The systematic uncertainties from experimental data include the efficiency corrections of single electron reconstruction and the dielectron pair reconstruction, which are estimated by methods similar to those described in refs. 37,38. The systematic uncertainties from the physics background contributions have three primary sources: the invariant yields, the branching ratios of hadrons that decay into $e^+e^-$, and the de-correlation effects on the mass distribution of the $c$ and $\bar{c}$ decayed dielectron pairs due to potential medium modifications. The uncertainties associated with the invariant yields of $\pi^0$, $\eta$, $\eta'$ are established based on previous experimental data[38,55,81] while the uncertainties for vector mesons ($\omega$, $\phi$ and $J/\psi$) are determined in this study.

The uncertainties in the $c\bar{c}$ yields are determined from two sources: the extrapolation of worldwide data and the values of $N_{\text{coll}}$. For the Drell-Yan production, the uncertainty is estimated by considering the uncertainties in PYTHIA simulations and in the values of $N_{\text{coll}}$. The $c\bar{c}$ pairs are largely produced as back-to-back pairs via the initial hard scatterings. However, interactions with the hot QCD medium at later stages can modify their kinematics, resulting in de-correlation effects on their decayed $e^+e^-$ pairs, which affects their reconstructed invariant mass distribution. To address these de-correlation effects, given the poorly known medium modifications, two extreme conditions are considered for systematic uncertainty estimation: (1) the angles of the single electron and positron are randomly assigned for their pair mass calculation, which effectively eliminates their correlations; (2) re-weight the $p_T$ of these $e^+$ and $e^-$ with the theoretical predictions from the Duke model[82] and the PHSD model[83,84] taking into account the strong interactions between charm quarks and the medium during the system evolution for the 200 GeV Au+Au collisions, where the medium modification effects are expected to be stronger than at 27 GeV and 54.4 GeV.

The systematic uncertainties on the thermal dielectron mass spectrum and the extracted temperatures are evaluated separately for individual sources by considering the variations with respect to their default values. The total systematic uncertainties are then determined by combining the individual uncertainties in quadrature. The primary source of systematic uncertainty on the fully corrected dielectron mass spectrum is due to uncertainties in the single electron reconstruction efficiencies. These uncertainties are ~ 7-10% for both the 27 and 54.4 GeV data sets, resulting in systematic uncertainties of roughly 10-20% and 40-45% for the thermal dielectron mass spectrum at 27 and 54.4 GeV, respectively. However, these uncertainties have minimal impact (1-2%) on the temperature extraction as they are largely correlated across mass bins and hence do not significantly distort the shape of the mass distribution. The dominant source of mass-dependent systematic uncertainty for the thermal dielectron mass spectrum is due to the simulation of cocktails. This leads to uncertainties of about 10-50% in the LMR and 10-20% in the IMR for the 27 GeV data, and 10-60% in the LMR and 10-30% in the IMR for the 54.4 GeV data. This source of uncertainty also dominates the systematic uncertainty of temperature measurements in the LMR at ~ 12-15%. The systematic uncertainty of temperatures from the IMR is found to be less than 10%, due to the relatively higher ratio of Data/CKTSum and smoother systematic uncertainties of the thermal dielectron mass spectrum.

## Centrality definition

In heavy-ion collisions, centrality is a physics quantity that quantifies the extent of overlap between the colliding nuclei. In this study, centrality is determined by aligning Monte Carlo Glauber simulations with the distributions of charged tracks in Au+Au collisions reconstructed in the STAR TPC, employing the methodologies described in ref. 80. The Au+Au collisions are classified into centrality intervals, presented as percentages of the total nucleus-nucleus inelastic interaction cross section. A smaller (larger) percentage corresponds to more central (peripheral) collisions.

## Data compared to theoretical models

Theoretical calculations from the TAMU model[7,8,26,85] and the PHSD model[23,24] have successfully reproduced previous thermal dilepton measurements in both SPS and RHIC heavy-ion collisions. As shown in the Supplementary Fig. 4 and Fig. 5, both models are compared to the spectra measured at STAR from 27 GeV and 54.4 GeV data in the different centralities. In general, both models can well describe the experimental data while the PHSD model seems to underestimate the data at high mass in the most peripheral Au+Au collisions (40-80% centrality) at 54.4 GeV. The generally steeper distributions in models compared to data primarily stem from the lower average temperatures assumed in the models.

## Temperatures at different centrality

To study the centrality dependence of the created thermal QCD medium, the associated thermal emission temperatures are extracted by fitting the thermal dielectron spectrum in different centralities. In these fits, the data are modeled using the integral of the fit function over each bin width. The extracted temperatures from both LMR and IMR are presented as a function of $N_{\text{part}}$, as shown in the Supplementary Fig. 6. In general, the temperatures at different centralities are consistent within uncertainties for both $T_{\text{LMR}}$ and $T_{\text{IMR}}$. Moreover, the temperatures from 27 GeV and 54.4 GeV Au+Au collisions are consistent for all centralities. Temperatures extracted from the LMR tend to cluster around the critical temperature from lattice QCD calculations. On the other hand, temperatures extracted in the IMR are generally higher than $T_{\text{LMR}}$ and $T_C$.

## Thermal lepton yield

The Supplementary Fig. 7 shows the thermal dilepton yields integrated over $0.4 < M_{ll} < 0.75$ GeV/$c^2$ as a function of the collision energy. These yields are divided by the average charged particle density to cancel out collision system volume effects. As the data show, the normalized thermal dilepton yields from the 0-80% centrality show no clear dependence on the collision energy. To directly prove that these thermal dileptons are emitted at an early stage of a collision when the system can be described as a hot QCD medium, the measured thermal dilepton yields are compared to the predictions from a statistical thermal model, as well as the $\rho^0 \rightarrow e^+e^-$ converted from the measured $\rho^0 \rightarrow \pi^+\pi^-$. The later is carried out with the data from $p+p$ collisions[48,86–88], $e^+ + e^-$ collisions[49,89,90] and peripheral Au+Au collisions[53] by considering the decay branching ratios of $\rho^0$ to $e^+e^-$ ($4.72 \times 10^{-5}$) and to $\pi^+\pi^-$ (~ 100%). In principle, these converted data can also represent the expected dielectron yields decayed from $\rho^0$ at chemical freeze-out. As one can see, the thermal dielectron yields are generally a factor of 5 higher than those expected from the chemical freeze-out. In order to investigate the impact of mass on the $\rho^0$ yield at chemical freeze-out, the statistical thermal model calculation with the $\rho^0$ pole mass altered to 0.4 GeV/$c^2$ is performed for this particular study and shown as a dashed line. As one can see, all the above comparisons provide solid evidence that the measured thermal dielectrons are predominantly from the stage when the collision system stays as a thermal hadronic/partonic source.

## Temperature extraction with thermal dimuon mass spectra from NA60 data

In 2009, the NA60 collaboration published the most precise thermal dilepton data for In+In collisions at $\sqrt{s_{\text{NN}}} = 17.3$ GeV. Here, we perform a temperature extraction based on the same method as described in the

main text using the NA60 collaboration data published in refs. 32,33,91. The fitting results are shown in the Supplementary Fig. 8. Note that the IMR data from Fig. 4.5 is used here, instead of that from Fig. 5.1 in ref. 33, because the former includes the systematic uncertainties. Temperature values from NA60 LMR and IMR data are extracted as $T_{LMR} = 172 \pm 6$ MeV and $T_{IMR} = 245 \pm 17$ MeV, respectively. We also note that the temperature is found to be $T_{LMR} = 151 \pm 2$ MeV when fitting to the LMR data from the NA60 Hard Probes conference proceedings[42].

## Data availability

The data for all STAR Collaboration papers are publicly available on the HEPData website: https://doi.org/10.17182/hepdata.147195. Data for this paper will be released following its publication.

## Code availability

The codes to process raw data collected by the STAR detector are publicly available on GitHub (https://github.com/star-bnl). The codes to analyze the produced data are not publicly available.

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

## Acknowledgements

We thank the RHIC Operations Group and SDCC at BNL, the NERSC Center at LBNL, and the Open Science Grid consortium for providing resources and support. This work was supported in part by the Office of Nuclear Physics within the U.S. DOE Office of Science, the U.S. National Science Foundation, National Natural Science Foundation of China,

Chinese Academy of Science, the Ministry of Science and Technology of China and the Chinese Ministry of Education, NSTC Taipei, the National Research Foundation of Korea, Czech Science Foundation and Ministry of Education, Youth and Sports of the Czech Republic, Hungarian National Research, Development and Innovation Office, New National Excellency Programme of the Hungarian Ministry of Human Capacities, Department of Atomic Energy and Department of Science and Technology of the Government of India, the National Science Centre and WUT ID-UB of Poland, the Ministry of Science, Education and Sports of the Republic of Croatia, German Bundesministerium für Bildung, Wissenschaft, Forschung and Technologie (BMBF), Helmholtz Association, Ministry of Education, Culture, Sports, Science, and Technology (MEXT), and Japan Society for the Promotion of Science (JSPS).

## Author contributions

All authors contributed to all research steps and writing of the paper.

## Competing interests

The authors declare no competing interests.

## Additional information

## STAR Collaboration

B. E. Aboona[1], J. Adam[2], L. Adamczyk[3], I. Aggarwal[4], M. M. Aggarwal[4], Z. Ahammed[5], A. K. Alshammri[6], E. C. Aschenauer[7], S. Aslam[8], J. Atchison[9], V. Bairathi[10], X. Bao[11], K. Barish[12], S. Behera[13], R. Bellwied[14], P. Bhagat[15], A. Bhasin[15], S. Bhatta[16], S. R. Bhosale[3], J. Bielcik[2], J. Bielcikova[2,17], J. D. Brandenburg[18], C. Broodo[14], X. Z. Cai[19], H. Caines[20], M. Calderón de la Barca Sánchez[21], D. Cebra[21], J. Ceska[2], I. Chakaberia[22], P. Chaloupka[2], Y. S. Chang[23], Z. Chang[24], A. Chatterjee[25], D. Chen[12], J. H. Chen[8], Q. Chen[26], Z. Chen[11], J. Cheng[27], Y. Cheng[28], W. Christie[7], X. Chu[7], S. Corey[18], H. J. Crawford[29], M. Csanád[30], G. Dale-Gau[31], A. Das[2], I. M. Deppner[32], A. Deshpande[16], A. Dhamija[4], A. Dimri[16], P. Dixit[33], X. Dong[22], J. L. Drachenberg[9], E. Duckworth[6], J. C. Dunlop[7], J. Engelage[29], G. Eppley[34], S. Esumi[35], O. Evdokimov[31], O. Eyser[7], R. Fatemi[36], S. Fazio[37], Y. Feng[23], E. Finch[38], Y. Fisyak[7], F. A. Flor[20], C. Fu[39], T. Fu[11], C. A. Gagliardi[1], T. Galatyuk[40], T. Gao[11], F. Geurts[34], A. Gibson[41], K. Gopal[13], X. Gou[11], D. Grosnick[41], A. Gu[42], A. Gupta[15], W. Guryn[7], A. Hamed[43], R. J. Hamilton[20], X. Han[18], Y. Han[34], S. Harabasz[40], M. D. Harasty[21], J. W. Harris[20], H. Harrison-Smith[36], L. B. Havener[20], X. H. He[39], Y. He[11], N. Herrmann[32], L. Holub[2], C. Hu[44], Q. Hu[39], Y. Hu[22], H. Huang[45,46], H. Z. Huang[28], S. L. Huang[16], T. Huang[31], Y. Huang[30], Y. Huang[47], M. Isshiki[35], W. W. Jacobs[24], A. Jalotra[15], C. Jena[13], A. Jentsch[7], Y. Ji[22], J. Jia[7,16], C. Jin[34], N. Jindal[18], X. Ju[48], E. G. Judd[29], S. Kabana[10], D. Kalinkin[36], K. Kang[27], D. Kapukchyan[12], K. Kauder[7], M. Kesler[6], A. Khanal[49], Y. V. Khyzhniak[18], D. P. Kikoła[50], J. Kim[7], D. Kincses[30], I. Kisel[51], A. Kiselev[7], A. G. Knospe[52], J. Kołaś[50], B. Korodi[18], L. K. Kosarzewski[18], L. Kumar[4], M. C. Labonte[21], R. Lacey[16], J. M. Landgraf[7], C. Larson[36], J. Lauret[7], A. Lebedev[7], J. H. Lee[7], Y. H. Leung[32], D. Li[48], H. -S. Li[23], H. Li[53], H. Li[26], W. Li[34], X. Li[48], X. Li[48], Y. Li[27], Z. Li[54], Z. Li[48], X. Liang[12], R. Licenik[2,17], T. Lin[11], Y. Lin[26], M. A. Lisa[18], C. Liu[39], G. Liu[54], H. Liu[42], L. Liu[47], Z. Liu[47], T. Ljubicic[34], O. Lomicky[2], R. S. Longacre[7], E. M. Loyd[12], T. Lu[39], J. Luo[48], X. F. Luo[47], L. Ma[8], R. Ma[7], Y. G. Ma[8], N. Magdy[55], D. Mallick[47], R. Manikandhan[14], S. Margetis[6], C. Markert[56], O. Matonoha[2], O. Mezhanska[2], K. Mi[47], S. Mioduszewski[1], B. Mohanty[57], B. Mondal[57], M. M. Mondal[57], I. Mooney[20], J. Mrazkova[2,17], M. I. Nagy[30], C. J. Naim[16], A. S. Nain[4], J. D. Nam[58], M. Nasim[33], H. Nasrulloh[48], J. M. Nelson[29], M. Nie[11], G. Nigmatkulov[31], T. Niida[35], T. Nonaka[35], G. Odyniec[22], A. Ogawa[7], S. Oh[59], K. Okubo[35], B. S. Page[7], S. Pal[2], A. Pandav[22], A. Panday[33], A. K. Pandey[39], T. Pani[60], A. Paul[12], S. Paul[16], D. Pawlowska[50], C. Perkins[29], J. Pluta[50], B. R. Pokhrel[58], I. D. Ponce Pinto[20], M. Posik[58], E. Pottebaum[20], S. Prodhan[13], T. L. Protzman[52], A. Prozorov[2], V. Prozorova[2], N. K. Pruthi[4], M. Przybycien[3], J. Putschke[49], Z. Qin[27], H. Qiu[39], C. Racz[12], S. K. Radhakrishnan[6], A. Rana[4], R. L. Ray[56], R. Reed[52], C. W. Robertson[23], M. Robotkova[2,17], M. A. Rosales Aguilar[36], D. Roy[60], P. Roy Chowdhury[50], L. Ruan[7], A. K. Sahoo[33], N. R. Sahoo[13], H. Sako[35], S. Salur[60], S. S. Sambyal[15], J. K. Sandhu[52], S. Sato[35], B. C. Schaefer[52], N. Schmitz[61], F. -J. Seck[40], J. Seger[62], R. Seto[12], P. Seyboth[61], N. Shah[63], P. V. Shanmuganathan[7], T. Shao[8], M. Sharma[15], N. Sharma[33], R. Sharma[13], S. R. Sharma[13],

A. I. Sheikh[6], D. Shen[11], D. Y. Shen[39], K. Shen[48], S. Shi[47], Y. Shi[11], F. Si[48], J. Singh[10], S. Singha[39], P. Sinha[13], M. J. Skoby[23,64], N. Smirnov[20], Y. Söhngen[32], Y. Song[20], T. D. S. Stanislaus[41], M. Stefaniak[18], Y. Su[48], M. Sumbera[17], X. Sun[39], Y. Sun[48], B. Surrow[58], M. Svoboda[2,17], Z. W. Sweger[21], A. C. Tamis[20], A. H. Tang[7], Z. Tang[48], T. Tarnowsky[65], J. H. Thomas[22], A. R. Timmins[14], D. Tlusty[62], D. Torres Valladares[34], S. Trentalange[28], P. Tribedy[7], S. K. Tripathy[50], T. Truhlar[2], B. A. Trzeciak[2], O. D. Tsai[7,28], C. Y. Tsang[6,7], Z. Tu[7], J. E. Tyler[1], T. Ullrich[7], D. G. Underwood[41,66], G. Van Buren[7], J. Vanek[7], I. Vassiliev[51], F. Videbæk[7], S. A. Voloshin[49], F. Wang[23], G. Wang[28], J. S. Wang[42], J. Wang[11], K. Wang[48], X. Wang[11], Y. Wang[48], Y. Wang[47], Y. Wang[27], Z. Wang[11], A. J. Watroba[3], J. C. Webb[7], P. C. Weidenkaff[32], G. D. Westfall[65], D. Wielanek[50], H. Wieman[22], G. Wilks[31], S. W. Wissink[24], R. Witt[67], C. P. Wong[7], J. Wu[44], X. Wu[28], X. Wu[48], B. Xi[8], Z. G. Xiao[27], G. Xie[44], W. Xie[23], H. Xu[42], N. Xu[47], Q. H. Xu[11], Y. Xu[11], Y. Xu[47], Z. Xu[6], Z. Xu[66], G. Yan[11], Z. Yan[16], C. Yang[11], Q. Yang[11], S. Yang[54], Y. Yang[45,46], Z. Ye[22], Z. Ye[54], L. Yi[11], Y. Yu[11], H. Zbroszczyk[50], W. Zha[48], C. Zhang[8], D. Zhang[54], J. Zhang[11], S. Zhang[68], W. Zhang[54], X. Zhang[39], Y. Zhang[39], Y. Zhang[48], Y. Zhang[11], Y. Zhang[26], Z. Zhang[7], Z. Zhang[31], F. Zhao[69], J. Zhao[8], S. Zhou[47], Y. Zhou[47], X. Zhu[27], M. Zurek[7,66] & M. Zyzak[51]

[1]Texas A&M University, College Station, TX, USA. [2]Czech Technical University in Prague, FNSPE, Prague, Czech Republic. [3]AGH University of Krakow, FPACS, Cracow, Poland. [4]Panjab University, Chandigarh, India. [5]Variable Energy Cyclotron Centre, Kolkata, India. [6]Kent State University, Kent, Ohio, USA. [7]Brookhaven National Laboratory, Upton, New York, USA. [8]Fudan University, Shanghai, China. [9]Abilene Christian University, Abilene, TX, USA. [10]Instituto de Alta Investigación, Universidad de Tarapacá, Arica, Chile. [11]Shandong University, Qingdao, Shandong, China. [12]University of California, Riverside, CA, USA. [13]Indian Institute of Science Education and Research (IISER) Tirupati, Tirupati, India. [14]University of Houston, Houston, TX, USA. [15]University of Jammu, Jammu, India. [16]State University of New York, Stony Brook, New York, USA. [17]Nuclear Physics Institute of the CAS, Rez, Czech Republic. [18]The Ohio State University, Columbus, OH, USA. [19]Shanghai Institute of Applied Physics, Chinese Academy of Sciences, Shanghai, China. [20]Yale University, New Haven, CT, USA. [21]University of California, Davis, CA, USA. [22]Lawrence Berkeley National Laboratory, Berkeley, CA, USA. [23]Purdue University, West Lafayette, IN, USA. [24]Indiana University, Bloomington, IN, USA. [25]National Institute of Technology Durgapur, Durgapur, India. [26]Guangxi Normal University, Guilin, China. [27]Tsinghua University, Beijing, China. [28]University of California, Los Angeles, CA, USA. [29]University of California, Berkeley, CA, USA. [30]ELTE Eötvös Loránd University, Budapest, Hungary. [31]University of Illinois at Chicago, Chicago, IL, USA. [32]University of Heidelberg, Heidelberg, Germany. [33]Indian Institute of Science Education and Research (IISER), Berhampur, India. [34]Rice University, Houston, TX, USA. [35]University of Tsukuba, Tsukuba, Ibaraki, Japan. [36]University of Kentucky, Lexington, KY, USA. [37]University of Calabria & INFN-Cosenza, Rende, Italy. [38]Southern Connecticut State University, New Haven, CT, USA. [39]Institute of Modern Physics, Chinese Academy of Sciences, Lanzhou, Gansu, China. [40]Technische Universität Darmstadt, Darmstadt, Germany. [41]Valparaiso University, Valparaiso, Indiana, USA. [42]Huzhou University, Huzhou, Zhejiang, China. [43]American University in Cairo, New Cairo, Egypt. [44]University of Chinese Academy of Sciences, Beijing, China. [45]Academia Sinica, Taipei, Taiwan. [46]National Cheng Kung University, Tainan, Taiwan. [47]Central China Normal University, Wuhan, Hubei, China. [48]University of Science and Technology of China, Hefei, Anhui, China. [49]Wayne State University, Detroit, MI, USA. [50]Warsaw University of Technology, Warsaw, Poland. [51]Frankfurt Institute for Advanced Studies FIAS, Frankfurt, Germany. [52]Lehigh University, Bethlehem, PA, USA. [53]Wuhan University of Science and Technology, Wuhan, Hubei, China. [54]South China Normal University, Guangzhou, Guangdong, China. [55]Texas Southern University, Houston, TX, USA. [56]University of Texas, Austin, TX, USA. [57]National Institute of Science Education and Research, HBNI, Jatni, India. [58]Temple University, Philadelphia, PA, USA. [59]Sejong University, Seoul, South Korea. [60]Rutgers University, Piscataway, NJ, USA. [61]Max-Planck-Institut für Physik, Munich, Germany. [62]Creighton University, Omaha, NE, USA. [63]Indian Institute Technology, Patna, Bihar, India. [64]Ball State University, Muncie, Indiana, USA. [65]Michigan State University, East Lansing, MI, USA. [66]Argonne National Laboratory, Argonne, IL, USA. [67]United States Naval Academy, Annapolis, MD, USA. [68]Chongqing University, Chongqing, China. [69]Lanzhou University, Lanzhou, China. ✉e-mail: star-publication@bnl.gov

