## [Transparent Peer Review file · Nature Communications]

Temperature Measurement of Quark-Gluon Plasma at Different Stages

Corresponding Author: Professor Zaochen Ye

Version 0:

Reviewer comments:

Reviewer #1

(Remarks to the Author)

Dear editor, the authors have revised their draft manuscript after my initial report, and that of two other referees. Here is my second report, after reading their revised manuscript.

There are several places in the text where somewhat vague assertions could be clarified. Examples are

- Page 2: "QGP consists of locally thermalized quarks and gluons...". This is technically incorrect according to our current state of knowledge. One finding from the RHIC and LHC programs is the modeling success of relativistic viscous fluid dynamics. By its very nature, a viscous fluid is not thermalized. In some limiting cases it may be "close to (kinetic) equilibrium", but not "locally thermalized" in any formal sense.
- Page 3: "In the early stage of QGP evolution, thermal dielectrons are predominantly produced via the annihilation processes among quarks and anti-quarks." This should be clarified or amended: In the low mass and intermediate mass regimes (as defined in this paper), next-to-leading order processes *also involving gluons* are the dominant channel in the dilepton production cross-section. See Fig. 2 of Phys. Rev. Lett. 64, 2242 (1990), or Fig. 6 of Phys. Rev. C 109, 044915 (2024). What the authors describe in the text is the LO (one-loop) contribution only, and the relative importance of (NLO+LO)/LO is invariant mass-dependent.
- Page 3, line 56: "teens of fm/c" -> "tens of fm/c"
- Page 5 line 68: the "intermediate-mass region" is invoked before its formal definition (lines 134-135, page 7).
- Page 9, line 169: "It should be noted that these temperature values from the IMR are systematically higher than model estimations using entropy and adiabatic thermodynamic expansions."
 - o It is not clear what "a model using entropy" is (ref. is to "the TAMU model")
 - o In the Methods section, Fig. 5 shows results with the same TAMU model: is this the model referred to earlier? What are the T values corresponding to the models shown in Fig. 5?
- The response of the authors to my previous comments are adequate. A closing remark: concerning the last point in my previous report about naming models and approaches. Indeed, there exists "many models named with (sic) a single scientist". That is exactly the point. The relevant papers referred to in this work are *not* single-author papers. I appreciate the other reviewers' concerns about data handling, significance, and interpretation. I will let them comment on the authors responses.

In what concerns points raised in this report, I can recommend publication after they are addressed.

Reviewer #2

(Remarks to the Author)

My comments are presented in the attached pdf file.

Reviewer #3

(Remarks to the Author)

Review of the paper

« Temperature Measurement of Quark-Gluon Plasma at Different Stages » by the STAR collaboration

I would like first to thank the authors for providing detailed answers to my comments and questions. I studied them in detail and read carefully the new version of the paper. Please find below my comments, which include in some places a reaction to the authors answers to my first review, and a few new comments, mostly following the outline of the draft:

Introduction:

I appreciate the changes to the text, which mainly concern the discussion of previous NA60 data, following the remarks of referee #2. They improve the readability of the paper and the reference to previous works.

Results:

Line 146: I do not understand the authors answer concerning the mass dependence of the rho width. In the BW function, is the (mass-dependent) total width, which, above the 2pion mass threshold, is dominated by the two-pion decay. So, the total width can be approximated by the 2pion decay width and the mass entering the sqrt function should be the pion mass? Anyway, the mass dependence of the dilepton decay width (following Vector Dominance Model) would be very different. Furthermore, as the electron mass is negligible w.r.t. M, the present expression leads to $\Gamma = 0$.

I appreciate that chi2 values have been added in fig.3. However, I also see that the two last points at 27 GeV are slightly higher than in the previous draft, albeit consistent with previous values within errors. What is the reason for this change? The dashed lines displaying the fit results are slightly different, as expected. However, I do not see any change in the IMR temperature values, which does not seem consistent to me.

In my first review I was questioning the consistency of doing two independent fits in the overlapping LMR and IMR region. Following my suggestion, the authors provided in their answer results of a global fit with two temperatures in the whole invariant mass range, which I find interesting. Of course, the IMR component is less well constrained in this way. On the other hand, with such a fit, the LMR component is not much affected, while higher IMR temperatures can be accessed. It directly shows that, keeping in mind a two-source approach, the average IMR temperature is affected by the low temperature stage. Probably such a global fit is not practicable for the STAR data and the authors have to live anyway with these average IMR temperatures, but for the NA60 data, wouldn't it give a better estimate of the QGP temperature?

Nevertheless, I understand that the same procedure should be applied for all data, for the sake of comparison, so I am fine with the way results are presented. Still, looking at the global fit results and considering the strong contribution of the LMR source in the overlap region and beyond, the statement in lines 168-169 "while those in the IMR are mainly from the earlier partonic stage with much higher temperatures" should be significantly smoothed.

An "average" temperature resulting from a fit of yields distributed in a given invariant mass range is in principle well defined and allows for a comparison of different systems and model predictions. However, I see a difficulty when the various sets of data have different distributions of errors in the different invariant mass bins.

For NA60 data, the IMR fit is very good and the uncertainties of the data points are of the same order of magnitude. So, the temperature can indeed be safely considered as an average temperature over the given mass interval. For STAR data however, despite the integration over very broad bins, the data points above 1.5 GeV/c² in the largest invariant mass bins of the interval have much larger uncertainties. One might therefore expect a lower sensitivity of the fits to the last points of the IMR invariant mass range. Indeed, the tests provided by the authors show a small sensitivity to the last point. But there is also a small sensitivity to the first point, so I cannot draw strong conclusions from this. Still the quality of the 54 GeV fit is worse and especially the large invariant mass region is not well described, so one could question the extraction of an "average" temperature in this case. I agree that, as the average temperature is expected to change depending on the invariant mass range, removing points at the edges of the interval is not a good test of the robustness of the fits, but rather of the sensitivity of the fit. Maybe additional tests could be done by changing the bin widths.

The data points are shown as an integrated yield over bins of variable widths, after division by bin width. The horizontal bar shows the width of the bin. This width is sometimes so wide, that the fit value is changing by a factor 7-10 from the lower to the upper edge of the bin. I guess the fits are based on the integral over the bin of the fit function and not on the value at the center of the bin, but this should be mentioned. In some bins only upper values are measured, how is this treated? As the results of the paper are based on the extraction of temperature parameters using fits of the data, it would be useful to indicate such important details in the method section.

It seems strange that the NA60 data are systematically lower than STAR data in the IMR region, while the temperatures are compatible within errors. Same remark for the comparison between STAR data at 54.4 and 27 GeV. Line 136-138: The authors stress that "the Au+Au IMR data are systematically higher than the NA60 data", which is indeed clear from fig. 3. Later they add "the thermal dileptons in the IMR may indicate sources with different temperatures in these three different energies". However, the fit results yield consistent values at all energies within errors. I see here a contradiction, which could be related to my remarks about the fits. In any case, such a contradiction should be clarified.

line 170 of the draft: It is mentioned that model estimations predict lower IMR values than measured ones. They also predict an increase of the IMR temperature with energy, which is not seen in the data. Is this just due to the large uncertainties or is there another possible reason?

lines 198-200: the interpretation of these overlapping temperatures is not clear. How do the new dilepton result shed light on the puzzle observed when comparing freeze-out temperatures and phase transition temperatures from LQCD? The authors mention the emission "over a long period of time at high density around a fixed temperature". The fact that this temperature is close to the phase transition one is an interesting result, which should be emphasized more, but it does not explain why the freeze-out temperature has a similar value and not a lower one. So, I do not think this conclusion of the paper is convincing enough.

Method:

I appreciate that Extended data fig. 1 has been added with information about the combinatorial background.

The authors write: "future experimental data with high statistics and further model studies are necessary". However, for the

LMR, the contribution of systematic errors to the temperature extraction is of the same order as the statistical error. Systematic errors are also important in the IMR region. So, is there some improvement to be expected in the direction of constraining Drell-Yan and semi-leptonic open charm hadron decays?

I read in lines 153-155: "In general, both models can well describe the experimental data while the PHSD model seems to underestimate the data from the most peripheral Au+Au collisions (40-80% centrality) at 54.4 GeV." However, I see in extended data fig. 5 that PHSD underestimate the data in this centrality region only in the high invariant mass region, and this effect is seen also at 27 GeV? There is a tendency towards steeper distribution in the models, hence smaller IMR temperatures. This could be mentioned here in relation with the remark at lines 169-171.

Detailed comments:

Line 56: for teens of MeV for tens of MeV

Line 68: theoretical line shapes from the TAMU model -> this concerns the rho meson ? I would write "theoretical rho lineshapes from the TAMU model"

Line 76 one finds "In the higher mass region (above 1 GeV/c²)," and then line 88: "There exists one intermediate-mass region measurement from NA60 37 at one beam energy to date." Intermediate -mass should be defined.

L88 of methods: Kt should be defined

In conclusion, I confirm the assessments made in my first review: there is no doubt about the interest of the scientific question addressed by the paper. The attempt to provide new measurements in an unexplored and very interesting region of the QCD phase space is very valuable. I am aware that these measurements are a unique chance to explore this region. However my concern is about the impact of the result for a broad community. The authors do their best to extract from their data the temperature of the produced medium in different dilepton invariant mass ranges. Nevertheless, the conclusions are limited by the precision of the signal extraction. The present measurements call for higher statistics, obviously not possible in a near future, but also for reducing systematic errors, by additional measurements or theoretical works. In my opinion, a Nature Communication reader is yearning for more. In addition, I still have some comments on the method. So, I would suggest to publish the letter, after revision, in a more specialized journal than Nature Communications.

Version 1:

Reviewer comments:

Reviewer #1

(Remarks to the Author)

Dear editor,

The authors have corrected most of the errors and clarified some of the imprecisions pointed out in my previous reports. However, I am somewhat puzzled by the bizarre insistence of the same authors in wanting to write that the quark gluon plasma (QGP) is "...locally thermalized" (page 2). This is quite simply incorrect. There are several ways to see this, owing to the success of viscous relativistic fluid mechanical simulations over the last two decades. First, writing the stress energy tensor in terms of particle fields, one needs a correction to the equilibrium distribution function. And that correction is popularly called " δf ". It can be calculated several ways, but its very existence quantifies the departure from equilibrium. Several sources are available for reference, one of them is Phys. Rev. C 81 (2010) 034907. Another way to see this is to note that the shear viscous tensor simply makes the stress energy tensor non-diagonal. In addition, the simple existence of a bulk viscosity (related to the non-conformal invariance of QCD in this context) implies a negative pressure. Therefore, the pressure in the cells is not that dictated by the equation of state, as many textbooks also show.

To conclude, I am somewhat baffled by the authors' argumentative stance and their persistence in stating something that would have been so very easy to correct. In addition, their curious answer to my original observation is even self-contradictory. First, in their response that questionable statement on page 2 is justified by a seemingly circular argument implying agreement of viscous hydrodynamics with data. Then, they write that "... locally thermalized refers to the condition where quarks and gluons ... achieve a near isotropic momentum distribution...". If equilibrium is achieved the momentum distribution *is* isotropic, not "near isotropic". Period. While this could appear as a nitpicking disagreement, such fine distinctions are expected from a scientific paper.

I will not comment on the observations made about the relevance and significance of the data – this is well done by others – but I leave to the editor to decide whether to publish a paper with seeming innocuous statements that are manifestly false.

Reviewer #3

(Remarks to the Author)

The authors made a significant effort to answer my critics, in particular, concerning the uncertainty of the fits, which I appreciate. Despite this progress, I still confirm my doubt about the impact of the results for a broad community.

Nevertheless, I accepted the editor request to review once again the paper and check the answers to my comments.

Most of the answers and text modifications are satisfactory, but there are still important pending issues, which are discussed in more details below.

Significance of the results: As already emphasized, I have no doubt that both the scientific question and the experimental approach are of strong interest for a broad community. The authors answer mainly concerns this aspect, but the impact of the new results is less convincing. In particular, as requested by reviewer #3, it should be more clearly indicated what is the novelty and the respective impact of STAR and Na60 data to the conclusions. One or two sentences in the conclusion could be enough for that, but the very short answer to reviewer #3 is not sufficient to me and I think his/her suggestions should be

taken into account and/or properly discussed.

Rho mass line-shape: I am still confused by the authors answer. The width entering the expression of the Breit-Wigner function (i.e. rho mass probability density) should obviously be the total width of the resonance, which is mainly the width of $\mu^+\mu^-$ decay. The expression for $\Gamma(M)$ in the paper corresponds to the decay of rho to two pions in the p wave (and not for the decay into two muons which would have a different mass behavior). Therefore $\Gamma(M)$ should be mentioned as being the total width and the pion mass should be used and not the muon mass. This has no significant numerical impact, but should be consistent. Let me also object that the form $(M^2 - 4m_{\pi}^2)^{3/2} / M$ is not just due to kinematics, it derives partly from kinematics and partly from the amplitude (p-wave decay to two pions).

Then the Boltzmann factor is introduced, which is fine, but I am missing the part related to the dimuon decay. It includes in principle a phase space factor and a factor due to the choice for the amplitude of the $\rho \rightarrow \gamma^*$. For example, in Eqs (1) and (2) of ref [41], the numerator in the dimuon yield expression is, as expected, the product of total width and the kinematical factors for the dimuon decay. These factors might not influence much the shape of the distribution, but please indicate your choice for the dimuon decay width in the text. B.t.w., Ref [41] has (to my humble opinion) the same error (muon mass instead of pion mass) in Eq.(2).

My remark on the statement "while those in the IMR are mainly from the earlier partonic stage with much higher temperatures" was related to the fact that the LMR contribution was extending in the IMR region. Looking at the additional "global fits" you provided in the previous round of review, the LMR contribution is dominant, at least up to 1.4 GeV/c².

Removing "much" in the sentence does not answer my remark. I would suggest "while those in the IMR are influenced by"

Version 2:

Reviewer comments:

Reviewer #3

(Remarks to the Author)

Please find below my final remarks and suggestions, which I consider as important for the consistency of the text, but should be easy to implement.

I then let the editor judge about the relevance of publishing this manuscript in Nature Comm., which, as mentioned previously would not have been my choice. This opinion is confirmed after these iterations.

The text about the Breit-Wigner expression improved, but should still be further clarified. Indeed, the expression for the rho mass line-shape is not directly the Breit-Wigner function, but the product of a Breit-Wigner function and the (in principle mass-dependent) branching ratio of $\rho \rightarrow \mu^+\mu^-$ ($Br = \Gamma_{II} / \Gamma$). It would be more consistent to have an expression of the type "the dilepton yield is proportional to ..."

Around the rho pole mass range, the kinematical factors are small, which is confirmed by your test using Na60 formula, which shows 2 MeV difference. So, this is fine.

However, the temperature values in the text changed by a few MeV (e.g. from 165 to 172 MeV in the case of Na60) w.r.t. the previous draft. The central values stay consistent with the range of the previous values including uncertainties, but the origin of this change should have been mentioned in your answer. Most importantly, the data point for NA60 LMR is still centered at 165 MeV in the plot!

Moreover, following the expression in the text Γ_{II} can not be identified to a width, I guess the pole width is missing, so please also correct that.

Finally, I am not sure if we expect any angular momentum dependence of the $\rho \rightarrow \mu^+\mu^-$ decay width, I suggest to simply remove "s-wave" in l 150.

REVIEWER COMMENTS

Reviewer #1 (Remarks to the Author):

Dear editor, the authors have revised their draft manuscript after my initial report, and that of two other referees. Here is my second report, after reading their revised manuscript.

There are several places in the text where somewhat vague assertions could be clarified. Examples are

- Page 2: “QGP consists of locally thermalized quarks and gluons...”. This is technically incorrect according to our current state of knowledge. One finding from the RHIC and LHC programs is the modeling success of relativistic viscous fluid dynamics. By its very nature, a viscous fluid is not thermalized. In some limiting cases it may be “close to (kinetic) equilibrium”, but not “locally thermalized” in any formal sense.

Response:

We respectfully disagree that a viscous fluid cannot be locally thermalized in the context of QGP. The success of relativistic viscous hydrodynamics in modeling RHIC and LHC data does not imply a lack of local thermalization. Rather, the QGP’s near-perfect fluid behavior—characterized by a shear viscosity-to-entropy density ratio (η/s) close to the theoretical limit of $1/(4\pi)$ —indicates a strongly coupled system that rapidly approaches local kinetic equilibrium. This is evidenced by the agreement between hydrodynamic predictions and experimental observables, such as elliptic flow (v_2) and particle spectra. The “locally thermalized” refers to the condition where quarks and gluons within small spatial domains achieve a near-isotropic momentum distribution, even as the system retains small viscous corrections. This is a prerequisite for the success of hydrodynamic theoretical works.

- Page 3: “In the early stage of QGP evolution, thermal dielectrons are predominantly produced via the annihilation processes among quarks and anti-quarks.” This should be clarified or amended: In the low mass and intermediate mass regimes (as defined in this paper), next-to-leading order processes *also involving gluons* are the dominant channel in the dilepton production cross-section. See Fig. 2 of Phys. Rev. Lett. 64, 2242 (1990), or Fig. 6 of Phys. Rev. C 109, 044915 (2024). What the authors describe in the text is the LO (one-loop) contribution only, and the relative importance of (NLO+LO)/LO is invariant mass-dependent.

Response: updated the sentence to be “In the early stage of QGP evolution, thermal dielectrons are predominantly produced via the annihilation processes among quarks and anti-quarks, and potentially gluons at higher order Refs[]”.

- Page 3, line 56: “teens of fm/c” -> “tens of fm/c”

Response: Changed as suggested.

- Page 5 line 68: the “intermediate-mass region” is invoked before its formal definition (lines 134-135, page 7).

Response: Intermediate-mass region ($m_{\{\phi\}} \lesssim M_{\{uu\}} \lesssim m_{\{J\psi\}}$)

- Page 9, line 169: “It should be noted that these temperature values from the IMR are systematically higher than model estimations using entropy and adiabatic thermodynamic expansions.” It is not clear what “a model using entropy” is (ref. is to “the TAMU model”)

Response: Done, added “the TAMU model”.

In the Methods section, Fig. 5 shows results with the same TAMU model: is this the model referred to earlier? What are the T values corresponding to the models shown in Fig. 5?

Response: Yes. In the TAMU model, the T_0 values for 27 and 54.4 GeV are around 250 MeV and 284 MeV, respectively, and it shows no dependency on centralities [Physics Letters B 753 (2016) 586–590].

- The responses of the authors to my previous comments are adequate. A closing remark: concerning the last point in my previous report about naming models and approaches. Indeed, there exists “many models named with (sic) a single scientist”. That is exactly the point. The relevant papers referred to in this work are *not* single-author papers.

I appreciate the other reviewers’ concerns about data handling, significance, and interpretation. I will let them comment on the authors’ responses.

In what concerns points raised in this report, I can recommend publication after they are addressed.

Response: We appreciate the referee’s positive feedback on our previous responses. We hope the answers above have adequately addressed the referee’s remaining comments. We sincerely thank you for your positive recommendation based on our satisfactory responses.

Reviewer #3 (Remarks to the Author):

Review of the paper

« Temperature Measurement of Quark-Gluon Plasma at Different Stages » by the STAR collaboration

I would like first to thank the authors for providing detailed answers to my comments and questions. I studied them in detail and read carefully the new version of the paper. Please find below my comments, which include in some places a reaction to the authors answers to my first review, and a few new comments, mostly following the outline of the draft:

Introduction:

I appreciate the changes to the text, which mainly concern the discussion of previous NA60 data, following the remarks of referee #2. They improve the readability of the paper and the reference to previous works.

Response:

We appreciate the referee's positive feedback to our modified manuscript. Please find our detailed responses embedded below.

Results:

Line 146: I do not understand the authors' answer concerning the mass dependence of the rho width. In the BW function, Γ is the (mass-dependent) total width, which, above the 2π mass threshold, is dominated by the two-pion decay. So, the total width can be approximated by the 2π decay width and the mass entering the sqrt function should be the pion mass? Anyway, the mass dependence of the dilepton decay width (following Vector Dominance Model) would be very different. Furthermore, as the electron mass is negligible w.r.t. M , the present expression leads to $\Gamma = 0$.

Response:

The term in the sqrt is purely due to decay kinematics and should be the mass of the daughters. The present expression leads to a constant decay width across rho mass, and does not lead to $\Gamma = 0$. Indeed the total decay width is dominated by the pion channel and therefore the pion mass term, but in the partial width, the daughter mass is the right mass term.

I appreciate that χ^2 values have been added in fig.3. However, I also see that the two last points at 27 GeV are slightly higher than in the previous draft, albeit consistent with previous values within errors. What is the reason for this change? The dashed lines displaying the fit results are slightly different, as expected. However, I do not see any change in the IMR temperature values, which does not seem consistent to me.

Response: Thanks for the good catch! We accidentally put a figure that corresponds to a special check to “re-scale the $c\bar{c}$ component” requested by Reviewer-2. Actually, our data has not changed. Now, the correct figure is used in the updated version.

In my first review I was questioning the consistency of doing two independent fits in the overlapping LMR and IMR region. Following my suggestion, the authors provided in their answer results of a global fit with two temperatures in the whole invariant mass range, which I find interesting. Of course, the IMR component is less well constrained in this way. On the other hand, with such a fit, the LMR component is not much affected, while higher IMR temperatures can be accessed. It directly shows that, keeping in mind a two-source approach, the average IMR temperature is affected by the low temperature stage. Probably such a global fit is not practicable for the STAR data and the authors have to live anyway with these average IMR temperatures, but for the Na60 data, wouldn't it give a better estimate of the QGP temperature?

Nevertheless, I understand that the same procedure should be applied for all data, for the sake of comparison, so I am fine with the way results are presented. Still, looking at the global fit results and considering the strong contribution of the LMR source in the overlap region and beyond, the statement in lines 168-169 “while those in the IMR are mainly from the earlier partonic stage with much higher temperatures” should be significantly smoothed.

Response: As we mentioned in the previous response, the average temperature is a well-defined quantity, but a slope at IMR after subtracting another component from the extrapolation of an average quantity (LMR) is not a well-defined quantity.

And we removed “much” from the sentence to smooth the statement.

An “average” temperature resulting from a fit of yields distributed in a given invariant mass range is in principle well defined and allows for a comparison of different systems and model predictions. However, I see a difficulty when the various sets of data have different distributions of errors in the different invariant mass bins.

For NA60 data, the IMR fit is very good and the uncertainties of the data points are of the same order of magnitude. So, the temperature can indeed be safely considered as an average temperature over the given mass interval. For STAR data however, despite

the integration over very broad bins, the data points above 1.5 GeV/c² in the largest invariant mass bins of the interval have much larger uncertainties. One might therefore expect a lower sensitivity of the fits to the last points of the IMR invariant mass range. Indeed, the tests provided by the authors show a small sensitivity to the last point. But there is also a small sensitivity to the first point, so I cannot draw strong conclusions from this. Still the quality of the 54 GeV fit is worse and especially the large invariant mass region is not well described, so one could question the extraction of an “average” temperature in this case. I agree that, as the average temperature is expected to change depending on the invariant mass range, removing points at the edges of the interval is not a good test of the robustness of the fits, but rather of the sensitivity of the fit. Maybe additional tests could be done by changing the bin widths. The data points are shown as an integrated yield over bins of variable widths, after division by bin width. The horizontal bar shows the width of the bin. This width is sometimes so wide, that the fit value is changing by a factor 7-10 from the lower to the upper edge of the bin. I guess the fits are based on the integral over the bin of the fit function and not on the value at the center of the bin, but this should be mentioned. In some bins only upper values are measured, how is this treated? As the results of the paper are based on the extraction of temperature parameters using fits of the data, it would be useful to indicate such important details in the method section.

Response: we fitted the data with the actual uncertainty for all bins (all data points have positive values), but plotted the data point as the downward arrows to denote that their statistical uncertainty is over 100%. Added a sentence “Downward arrows indicate statistical uncertainties exceeding 100%.” in the caption to clarify on this.

The previous fits were performed based on the data at the bin center, now we performed the fit with the integral over the bins, and found the extracted temperature values are well consistent between these two cases as shown the following two figures:

Considering that the fit is performed with wide-bin data, the results fitted with “integral over the bin” should be a better option. So we updated our results in both figures and texts with the temperatures fitted using the “integral over bin”. And added one sentence in the method part line159-162 to clarify on this as follows:

“To study the centrality dependence of the created thermal QCD medium, the associated thermal emission temperatures are extracted by fitting the thermal dielectron spectrum in different centralities. In these fits, the data are modeled using the integral of the fit function over each bin width.”

It seems strange that the NA60 data are systematically lower than STAR data in the IMR region, while the temperatures are compatible within errors. Same remark for the comparison between STAR data at 54.4 and 27 GeV. Line 136-138: The authors stress that “the Au+Au IMR data are systematically higher than the NA60 data”, which is indeed clear from fig. 3. Later they add “the thermal dileptons in the IMR may indicate sources with different temperatures in these three different energies”. However, the fit results yield consistent values at all energies within errors. I see a contradiction here, which could be related to my remarks about the fits. In any case, such a contradiction should be clarified.

Response: Because STAR data are at higher energies with larger collision systems (Au+Au vs In+In), it is possible that our dilepton yields are higher due to system size while the temperature is comparable. We also note that the temperatures at higher energies are still systematically higher than NA60, although they are indeed comparable within errors.

We also slightly rephrase the sentence to be “the Au+Au IMR central values are systematically higher than the NA60 data”.

line 170 of the draft: It is mentioned that model estimations predict lower IMR values than measured ones. They also predict an increase of the IMR temperature with energy, which is not seen in the data. Is this just due to the large uncertainties or is there another possible reason?

Response: Indeed, the model predicts an approximately 14% higher temperature for QGP in Au+Au collisions at 54.4 GeV compared to 27 GeV. However, the precision of our current data is insufficient to discern a clear increasing trend in the extracted IMR temperature with collision energy. To clarify on this, we added one sentence in the draft: “Although theoretical studies Refs[14,16,30,31] anticipate a hotter QGP created at higher collision energies, this trend is not apparent in our data due to the current precision limitations.”

lines 198-200: the interpretation of these overlapping temperatures is not clear. How do the new dilepton results shed light on the puzzle observed when comparing freeze-out temperatures and phase transition temperatures from LQCD? The authors mention the

emission “over a long period of time at high density around a fixed temperature”. The fact that this temperature is close to the phase transition one is an interesting result, which should be emphasized more, but it does not explain why the freeze-out temperature has a similar value and not a lower one. So, I do not think this conclusion of the paper is convincing enough.

Response: We believe that the manuscript’s phrasing is consistent with the Referee’s statements. Our data reveal an average emission temperature from the dilepton measurement that closely matches both the LQCD phase transition temperature and chemical freeze-out temperature determined from hadron yields. While the underlying reason remains unexplained, we view this as an intriguing experimental finding that could pave ways for the future theoretical efforts toward a comprehensive understanding.

Method:

I appreciate that Extended data fig. 1 has been added with information about the combinatorial background.

The authors write: “future experimental data with high statistics and further model studies are necessary”. However, for the LMR, the contribution of systematic errors to the temperature extraction is of the same order as the statistical error. Systematic errors are also important in the IMR region. So, is there some improvement to be expected in the direction of constraining Drell-Yan and semi-leptonic open charm hadron decays?

Response: For the heavy-ion collisions at these energies, our measurements are the final results, as no ongoing or planned experiments can repeat on these in the next decade.

However, there were about 8 billion Au+Au 200 GeV collision data collected in 2023, and this year RHIC run we plan to collect more and target a total of 20 billion Au+Au collision events at 200 GeV combining years 2023 and 2025. These will allow a better determination of hadron Dalitz decays which dominate the systematic uncertainties of LMR measurement. The huge datasets also offer potential reductions in both statistical and systematic uncertainties for the LMR and IMR measurements at RHIC top energies. Meanwhile, future experiments at LHC and SPS energies may achieve better precision, benefiting from both large datasets and better control on open charm contribution with the displayed vertex reconstruction by silicon detectors.

The sentence of “future experimental data with high statistics and further model studies are necessary” is updated to “Future experiments with enhanced statistical and systematic precision, alongside further theoretical studies, are essential to clarify these observations.”

I read in lines 153-155: “In general, both models can well describe the experimental data while the PHSD model seems to underestimate the data from the most peripheral Au+Au collisions (40-80% centrality) at 54.4 GeV.” However, I see in extended data fig. 5 that PHSD underestimate the data in this centrality region only in the high invariant mass region, and this effect is seen also at 27 GeV? There is a tendency towards steeper distribution in the models, hence smaller IMR temperatures. This could be mentioned here in relation with the remark at lines 169-171.

Response: Updated the sentence to be “In general, both models can well describe the experimental data while the PHSD model seems to underestimate the data at high mass and in the most peripheral Au+Au collisions (40-80% centrality) at 54.4 GeV.” And added one more sentence “The generally steeper distributions in models compared to data primarily stem from the lower average temperatures assumed in the models.”

Detailed comments:

Line 56: for teens of MeV for tens of MeV

Response: This is updated as suggested.

Line 68: theoretical line shapes from the TAMU model -> this concerns the rho meson ? I would write “theoretical rho lineshapes from the TAMU model”

Response: actually it is the total lineshape (rho+QGP). To better clarify it, the sentence is modified to be “theoretical lineshape of total thermal dielectrons from the TAMU model”.

Line 76 one finds “In the higher mass region (above 1 GeV/c²),” and then line 88: “There exists one intermediate-mass region measurement from NA60 37 at one beam energy to date.” Intermediate -mass should be defined.

Response: updated the sentence to be “Intermediate-mass region ($m_{\phi} \sim M_{\{uu\}} \sim m_{\psi}$)”.

L88 of methods: k_T should be defined

Response: Done. Defined as “the primordial k_T (the intrinsic transverse momentum that partons of the colliding protons have before the hard scattering process occurs, which controls the p_T distribution of final-state particles)”

In conclusion, I confirm the assessments made in my first review: there is no doubt about the interest of the scientific question addressed by the paper. The attempt to provide new measurements in an unexplored and very interesting region of the QCD

phase space is very valuable. I am aware that these measurements are a unique chance to explore this region. However my concern is about the impact of the result for a broad community. The authors do their best to extract from their data the temperature of the produced medium in different dilepton invariant mass ranges. Nevertheless, the conclusions are limited by the precision of the signal extraction. The present measurements call for higher statistics, obviously not possible in a near future, but also for reducing systematic errors, by additional measurements or theoretical works. In my opinion, a Nature Communication reader is yearning for more. In addition, I still have some comments on the method. So, I would suggest to publish the letter, after revision, in a more specialized journal than Nature Communications.

Response: We sincerely thank the referee for acknowledging the scientific merit and uniqueness of our measurements exploring an uncharted region of QCD phase space. While we recognize reviewer's concerns about the broad impact and precision limitations of signal extraction, we believe our results hold significant interdisciplinary appeal and relevance, as outlined below, justifying their fit for Nature Communications.

Interdisciplinary Significance:

Thermal radiation studies span multiple disciplines, as all matter radiates and its spectra reflect its temperature, and our work probes the QGP—a state matter of under extreme conditions (even quarks and gluons are deconfined)—via thermal dilepton pairs. The invariant mass spectra of dileptons offer a Lorentz-invariant “thermometer” across the QGP’s evolution, from early stages to its phase transition. By studying the Au+Au collisions at varying energies, we explore the QCD phase diagram with diverse trajectories in terms of temperature and baryon chemical potential. This study, for the first time, reports the average QGP temperatures at two evolutionary stages and at different collision energies. The findings in this study not only refines our understanding of QGP thermodynamics but also informs broader fields like astrophysics and plasma physics, where thermal emission under extreme conditions is key.

Broader Appeal: Measuring the temperature of matter under extreme conditions, such as the QGP, which is predicted to be formed microseconds after the Big Bang, captivates both scientific and general audiences, akin to using an infrared thermometer to gauge a distant object’s heat. Our approach—leveraging a massive detector system and advanced algorithms to extract trillion-degree temperatures from thermal dileptons emitted by relativistic thermalized quarks and gluons—offers a compelling narrative of cutting-edge science, accessible to Nature Communications’ diverse readership.

Reviewer #2 (Remarks to the Author):

My comments are presented in the attached pdf file.

General comments

Most of the specific technical questions that I asked in the previous iteration have been answered. However, my general criticism of the paper remains. The STAR data suffer from statistical limitations due to the lower luminosity available at the RHIC collider compared to fixed-target experiments, such as HADES and NA60. There are also limitations in the experimental setup, such as the inability to tag charm. Poor statistics and the impossibility to tag charm penalize the IMR more than the LMR, making it difficult to draw quantitative conclusions using the STAR data alone.

The paper reports that the temperature extracted from STAR data in the low-mass region (LMR), averaged over all energies, is $(1.99 \pm 0.24) \times 10^{12}$ K, with a 12% error. In contrast, the temperature extracted from NA60 data (ref. Eur. Phys. J. C59 (2009) 607) is 165 ± 4 MeV, with a relative error of 2.4%. If one uses the NA60 data from AIP Conf. Proc. 1322 (2010) 1, which are based on more accurate mass calibrations and acceptance corrections, the quoted temperature is 151 ± 2 MeV, with a relative error of 1.3%. This is 5–10 times smaller than the STAR averaged data.

In the intermediate-mass region (IMR), the temperature from STAR data, averaged over all energies, is $(3.4 \pm 0.55) \times 10^{12}$ K, with a 16% error. The original measurement by NA60 (AIP Conf. Proc. 1322 (2010) 1) is 205 ± 12 MeV, with a relative error of 6%, while the fit quoted in the present paper, in a different mass range, is 245 ± 17 MeV, with a relative error of 7%. This is ~ 2.5 times more precise than the averaged STAR data.

The average temperatures in the LMR and IMR from STAR data are compatible within 1.8σ . Not averaging over all data, STAR data span different baryochemical potential (μ_B) values, some of which were not measured so far. This is, of course, interesting. On the other hand, in this case, the IMR temperatures are so large that they are compatible with the LMR temperatures within 1.5σ or less.

Therefore, the main conclusions of the paper—based on a quantitatively larger temperature in the IMR compared to the LMR, signaling different sources of thermal radiation in the two regions—rest mainly on the NA60 data. It is not clear what kind of quantitative advancement we could achieve, such as improving our present knowledge of the equation of state, in a regime

of moderate μ_B , using the STAR data alone or adding them to the existing NA60 data.

The authors should explicitly acknowledge these limitations in the manuscript and discuss how they impact the interpretation of the results. They should provide a more detailed discussion of why the STAR data, despite their limitations, might still be valuable. However, they should also clearly state that the NA60 data remain the benchmark for precision and that the STAR results are complementary rather than decisive.

At the theoretical level, the microscopic many-body models, such as the one by Ralf Rapp and collaborators, developed during the 1990s, were very successful in describing the NA60 data, aligning with the main message of the present paper. NA60 data were already found to be consistent with radiation produced in the quark-gluon plasma and radiation from the hadronic phase dominated by the strongly broadened ρ meson, separated by a phase transition. The paper Nucl. Phys. A 806, 339 (2008), originally cited in the NA60 paper Eur. Phys. J. C 59 (2009) (fig. 5.1), describes at length the model and also different scenarios for the equation of state and their implications for the radiation from the partonic phase versus radiation from the hadronic phase. The more recent paper Phys. Lett. B 753 (2016) 586 makes use of a modern lattice QCD equation of state and shows that the predictions of hadronic many-body theory for a *melting* ρ meson, coupled with quark-gluon plasma emission, yield a quantitative description of the dilepton spectra at the SPS (see fig. 1).

Furthermore, as I pointed out in the previous iteration, it is an established fact since long time that the radiation from the LMR shines mostly from the medium at temperatures very close to T_c . This is reported, for instance, in J. Phys. Conf. Ser. 420 (2013) 012017 by R. Rapp, and also in Acta Phys. Polon. B 42 (2011) 2823, using the full-fledged theoretical framework published in referred journals to describe the NA60 data and cited also in the present paper.

The introduction part of the present paper provides an overview of thermal radiation produced by real and virtual photons in the hot medium produced in a high-energy nuclear collision and cites past measurements. However, it is thin in the discussion of the results in terms of their meaning from the comparison of the data to the theory along the lines discussed in the previous paragraphs (though there are references to many theoretical papers). The ρ broadening is mentioned almost in passing without a solid discussion of its

implications. To state again things in a slightly different way, the observed broadening in the NA60 data exceeds 400 MeV, approaching a divergent width—a *melting* ρ —as expected close to T_c in an almost chirally restored medium. On the other hand, this seems a novel result obtained for the first time from this analysis, while reading the section on results and discussion.

The authors should expand the discussion in the introduction and results sections to link more explicitly the findings to the existing theoretical models. References to key theoretical works, such as those by Ralf Rapp and collaborators, should be integrated more thoroughly.

Summarizing, I do not recommend the paper for publication in *Nature Communications Physics* because it does not present new results that change or improve the paradigm for the interpretation of thermal radiation produced in high-energy heavy ion collisions, which has been consolidated for the past two decades. The paper is worth publishing in more specialized journals, provided the introduction and discussion of results are revised as indicated above.

Specific comments

NA60 data are used in the present paper, and the authors remarked that it is unclear what changed from Eur. Phys. J. C59 (2009) 607 to AIP Conf. Proc. 1322 (2010) 1. I checked existing literature from NA60. First, there was an improvement in the acceptance correction procedure, which was originally multi-differential in Eur. Phys. J. C59 (2009) 607. Then, in order to improve the statistical accuracy, it was shown that it could be reduced to a p_T - M correction by integrating over other kinematic variables. I found this described in Nucl. Phys. A 783 (2007) 327.

The other aspect is the subtraction of the narrow resonances, which I commented on already in the previous iteration. An improved mass calibration allowed the narrow structures to be subtracted more precisely, leaving a smoother continuum. I found this described in Eur. Phys. J. C64 (2009) (NA60 paper on ϕ meson production). This procedure was applied to the whole LMR in AIP Conf. Proc. 1322, 1–10 (2010).

Response: We have already stated that “NA60 experiment reported a benchmark result ...” in lines 76-80. The reviewer refers to conference proceedings as a benchmark,

which are public but after more than a decade have never been submitted for peer review. **[editorial note: redacted for confidentiality]**

REVIEWER COMMENTS

Reviewer #1 (Remarks to the Author):

Dear editor,

The authors have corrected most of the errors and clarified some of the imprecisions pointed out in my previous reports. However, I am somewhat puzzled by the bizarre insistence of the same authors in wanting to write that the quark gluon plasma (QGP) is "...locally thermalized" (page 2). This is quite simply incorrect. There are several ways to see this, owing to the success of viscous relativistic fluid mechanical simulations over the last two decades. First, writing the stress energy tensor in terms of particle fields, one needs a correction to the equilibrium distribution function. And that correction is popularly called " δf ". It can be calculated several ways, but its very existence quantifies the departure from equilibrium. Several sources are available for reference, one of them is Phys. Rev. C 81 (2010) 034907. Another way to see this is to note that the shear viscous tensor simply makes the stress energy tensor non-diagonal. In addition, the simple existence of a bulk viscosity (related to the non-conformal invariance of

QCD in this context) implies a negative pressure. Therefore, the pressure in the cells is not that dictated by the equation of state, as many textbooks also show.

To conclude, I am somewhat baffled by the authors' argumentative stance and their persistence in stating something that would have been so very easy to correct. In addition, their curious answer to my original observation is even self-contradictory. First, in their response that questionable statement on page 2 is justified by a seemingly circular argument implying agreement of viscous hydrodynamics with data. Then, they write that "... locally thermalized refers to the condition where quarks and gluons ... achieve a near isotropic momentum distribution...". If equilibrium is achieved the momentum distribution *is* isotropic, not "near isotropic". Period. While this could appear as a nitpicking disagreement, such fine distinctions are expected from a scientific paper.

I will not comment on the observations made about the relevance and significance of the data – this is well done by others – but I leave it to the editor to decide whether to publish a paper with seemingly innocuous statements that are manifestly false.

Response:

We sincerely appreciate the reviewer's detailed comments. In response, we have removed the "locally thermalized" from the sentence, and updated it to "QGP consists of quarks and gluons with temperatures in excess of hundreds of MeV" in the revised manuscript.

Reviewer #3 (Remarks to the Author):

The authors made a significant effort to answer my critics, in particular, concerning the uncertainty of the fits, which I appreciate. Despite this progress, I still confirm my doubt about the impact of the results for a broad community. Nevertheless, I accepted the editor's request to review once again the paper and check the answers to my comments.

Most of the answers and text modifications are satisfactory, but there are still important pending issues, which are discussed in more details below. Significance of the results: As already emphasized, I have no doubt that both the scientific question and the experimental approach are of strong interest for a broad community. The author's answer mainly concerns this aspect, but the impact of the new results is less convincing. In particular, as requested by reviewer #3, it should be more clearly indicated what is the novelty and the respective impact of STAR and Na60 data to the conclusions. One or two sentences in the conclusion could be enough for that, but the very short answer to reviewer #3 is not sufficient to me and I think his/her suggestions should be taken into account and/or properly discussed.

Response:

We appreciate reviewer's acknowledgement of the scientific question and experimental approach of this study, as well as the previous detailed comments which significantly improved the quality of this manuscript.

To further describe the novelty and the respective impact of STAR and NA60 data, we have added the following sentences as the last paragraph:

“Thermal dileptons were proposed to serve as a critical thermometer of QGP created in high-energy heavy-ion collisions. However, their production rate is very low and the physics background is large. The extraction of temperature from the dilepton spectra has been limited to a single beam energy in a relatively small collision system performed by the NA60 experiment. By studying Au+Au collisions at various energies, we explore the QCD phase diagram across different temperatures and baryon chemical potentials. For the first time, we report the average QGP temperatures at two stages of their time evolution and at multiple baryonic chemical potentials, advancing our understanding of QGP thermodynamics.”

Rho mass line-shape: I am still confused by the authors' answer. The width entering the expression of the Breit-Wigner function (i.e. rho mass probability density) should obviously be the total width of the resonance, which is mainly the width of $\mu^+\mu^-$ decay.

The expression for $\Gamma(M)$ in the paper corresponds to the decay of rho to two pions in the p wave (and not for the decay into two muons which would have a different mass behavior). Therefore $\Gamma(M)$ should be mentioned as being the total width and the pion mass should be used and not the muon mass. This has no significant numerical impact, but should be consistent. Let me also object that the form $(M^2 - 4m_\pi^2)^{3/2} / M$ is not just due to kinematics, it derives partly from kinematics and partly from the amplitude (p-wave decay to two pions). Then the Boltzmann factor is introduced, which is fine, but I am missing the part related to the dimuon decay. It includes in principle a phase space factor and a factor due to the choice for the amplitude of the $\rho \rightarrow \mu^+\mu^-$. For example, in Eqs (1) and (2) of ref [41], the numerator in the dimuon yield expression is, as expected, the product of total width and the kinematical factors for the dimuon decay. These factors might not influence much the shape of the distribution, but please indicate your choice for the dimuon decay width in the text. B.t.w., Ref [41] has (to my humble opinion) the same error (muon mass instead of pion mass) in Eq.(2).

Response:

Thanks for these very good comments! We agree with your comments. We have updated the fit function for the T_{LMR} extraction and updated the sentences in the revised manuscript to explain every term more clearly. As you expected, it does not make any visible difference to the extracted T_{LMR} values in Fig.4.

Please find the updated sentences describing the fit functions as follows:

146 the continuum thermal distribution is used to fit the measured mass spectrum. The mass line-
147 shape of ρ^0 decaying to dileptons in vacuum can be described by a relativistic Breit-Wigner func-
148 tion⁵⁴⁻⁵⁶, $f^{\text{BW}}(M) = \frac{MM_0\Gamma_u}{(M_0^2 - M^2)^2 + M_0^2\Gamma^2}$, where M is the invariant mass of the dilepton pair,
149 $\Gamma = \Gamma_0 \frac{M_0}{M} \left(\frac{M^2 - 4m_\pi^2}{M_0^2 - 4m_\pi^2}\right)^{3/2}$ is the total width, predominantly influenced by the p -wave decay
150 $\rho^0 \rightarrow \pi^+\pi^-$, and $\Gamma_u = \left(1 + \frac{2m_l^2}{M^2}\right) (1 - 4m_l^2/M^2)^{1/2}$ denotes the s -wave decay width^{57,58} for
151 $\rho^0 \rightarrow l^+l^-$. Here, M_0 and Γ_0 are the pole mass and width of ρ^0 meson, while m_l is the lepton mass.
152 When present inside a hot QCD medium, the ρ^0 mass lineshape can be described by $f^{\text{BW}}(M)$,
153 multiplied by $M^{3/2}e^{-M/k_{\text{B}}T}$ to account for thermal radiation rate and phase space^{40,41,59,60}. Both
154 $f^{\text{BW}}(M)$ and the Boltzmann factor ($e^{-M/k_{\text{B}}T}$) are highly dependent on the medium temperature.
155 If the ρ^0 is completely dissolved in the medium, its mass spectral structure ($f^{\text{BW}}(M)$) spreads
156 out and approaches a smooth distribution similar to a $q\bar{q}$ continuum (QGP thermal radiation)^{14,36}

We note that several different functions related to the line shapes of $\rho \rightarrow l^+l^-$ were discussed in previous publications [PLB 757 (2016) 437-444, PRC 107, 025208 (2023), PRD 104, 034021 (2021), NPA 582 (1995) 731-748, PRC 94, 054905 (2016)]. While these studies explore different functional forms, they do not provide a definitive consensus on the optimal line shape. In this work, we selected the fit function described in the text because it best captures the physics processes relevant to this study, explained in the above texts in the paper. The chosen line shape aligns closely with that used in the NA60 paper [PLB 757 (2016) 437-444], which provides the most directly relevant reference for the $\rho \rightarrow l^+l^-$ line shape.

As discussed, we agree that Γ_ρ represents the total width of the ρ meson. Consequently, we adopted m_π instead of m_μ in the $(\dots)^{3/2}$ term to reflect the dominant $\rho \rightarrow \pi^+\pi^-$ decay width. Additionally, our line shape differs from that of the NA60 paper in the numerator. While we employ the conventional relativistic Breit-Wigner form with Γ_{ll} in the numerator [PRL 89, 272302 (2002), PRL 92 092301 (2004)], the NA60 formulation includes an additional factor, $(1 - (m_\pi^{3/2}/M)^2)^{3/2}$, corresponding to the vacuum $\rho \rightarrow \pi^+\pi^-$ decay. The NA60 formulation is as follows:

$$\frac{dN}{dM} \propto \frac{\sqrt{1 - \frac{4m_\pi^2}{M^2}} \left(1 + \frac{2m_l^2}{M^2}\right) \left(1 - \frac{4m_l^2}{M^2}\right)^{3/2}}{\left(m_\rho^2 - M^2\right)^2 + m_\rho^2\Gamma_\rho^2(M)} (MT)^{3/2} e^{-\frac{M}{T_\rho}} \quad (2)$$

$$\Gamma_{\rho}(M) = \Gamma_{0\rho} \frac{m_{\rho}}{M} \left(\frac{M^2/4 - m_{\mu}^2}{m_{\rho}^2/4 - m_{\mu}^2} \right)^{3/2} = \Gamma_{0\rho} \frac{m_{\rho}}{M} \left(\frac{q}{q_0} \right)^3. \quad (3)$$

To evaluate the numerical effects, we checked the performance of the T_LMR extraction using the fit function described in this paper (left) and the fit function described in the NA60 paper (right) on the NA60 LMR data below, the fit results are well consistent (only 2 MeV difference) between these two cases.

We also changed the m_{π} to m_{μ} in the total width Γ in the fit, and confirmed that there is very little numerical effect on the T_LMR extraction (only changed by 1 MeV).

My remark on the statement “while those in the IMR are mainly from the earlier partonic stage with much higher temperatures” was related to the fact that the LMR contribution was extending in the IMR region. Looking at the additional “global fits” you provided in the previous round of review, the LMR contribution is dominant, at least up to 1.4 GeV/c². Removing “much” in the sentence does not answer my remark. I would suggest “while those in the IMR are influenced by

Response:

Agree. Updated the sentence “while those in the IMR are mainly from” to be “while those in the IMR are influenced by”.

Responses are in blue.

REVIEWERS' COMMENTS

Reviewer #3 (Remarks to the Author):

Please find below my final remarks and suggestions, which I consider as important for the consistency of the text, but should be easy to implement.

I then let the editor judge about the relevance of publishing this manuscript in Nature Comm., which, as mentioned previously, would not have been my choice. This opinion is confirmed after these iterations.

The text about the Breit-Wigner expression improved, but should still be further clarified. Indeed, the expression for the rho mass line-shape is not directly the Breit-Wigner function, but the product of a Breit-Wigner function and the (in principle mass-dependent) branching ratio of rho -> mu+ mu- ($Br = \Gamma_{II} / \Gamma$). It would be more consistent to have an expression of the type “the dilepton yield is proportional to ...”

Response: Updated the sentence of “When present inside a hot QCD medium, the ρ^0 mass lineshape can be described by $f^{\text{BW}}(M)$, multiplied by $M^{\{3/2\}} e^{\{-M/kBT\}}$ to account for thermal radiation rate and phase space” to be “The dilepton yields from these in-medium ρ^0 meson decays are proportional to the $f^{\text{BW}}(M)$ multiplied by $M^{\{3/2\}} e^{\{-M/kBT\}}$.”

Around the rho pole mass range, the kinematical factors are small, which is confirmed by your test using Na60 formula, which shows 2 MeV difference. So, this is fine. However, the temperature values in the text changed by a few MeV (e.g. from 165 to 172 MeV in the case of Na60) w.r.t. the previous draft. The central values stay consistent with the range of the previous values including uncertainties, but the origin of this change should have been mentioned in your answer. Most importantly, the data point for NA60 LMR is still centered at 165 MeV in the plot!

Response: Yes, the T_{LMR} of NA60 is updated from 165 +/- 4 MeV to 172 +/- 6 MeV (well consistent within uncertainties) due to the updated fitting curve, as indicated in the previously submitted diff file. We regret not directly mentioning it in the previous response file. Thanks for

$\left(1 + \frac{2m_i^2}{M^2}\right) (1 - 4m_i^2/M^2)^{1/2}$., The NA60 LMR data point is updated in Fig.4.

Moreover, following the expression in the text Γ_{II} can not be identified to a width. I guess Finally, I am not sure if we expect any angular momentum dependence of the $\rho^0 \rightarrow \mu^+ \mu^-$ decay width, I suggest simply removing “s-wave” in l 150.

Response: Yes, the expression for Γ_{II} represents only the mass-dependent component. Response: Removed “s-wave”.

To better clarify on this, the sentence is updated to be: “the $\rho^0 \rightarrow l+l^-$ decay width $\Gamma_{\text{II}} \propto$

Referee report

General comments

Most of the specific technical questions that I asked in the previous iteration have been answered. However, my general criticism of the paper remains. The STAR data suffer from statistical limitations due to the lower luminosity available at the RHIC collider compared to fixed-target experiments, such as HADES and NA60. There are also limitations in the experimental setup, such as the inability to tag charm. Poor statistics and the impossibility to tag charm penalize the IMR more than the LMR, making it difficult to draw quantitative conclusions using the STAR data alone.

The paper reports that the temperature extracted from STAR data in the low-mass region (LMR), averaged over all energies, is $(1.99 \pm 0.24) \times 10^{12}$ K, with a 12% error. In contrast, the temperature extracted from NA60 data (ref. Eur. Phys. J. C59 (2009) 607) is 165 ± 4 MeV, with a relative error of 2.4%. If one uses the NA60 data from AIP Conf. Proc. 1322 (2010) 1, which are based on more accurate mass calibrations and acceptance corrections, the quoted temperature is 151 ± 2 MeV, with a relative error of 1.3%. This is 5–10 times smaller than the STAR averaged data.

In the intermediate-mass region (IMR), the temperature from STAR data, averaged over all energies, is $(3.4 \pm 0.55) \times 10^{12}$ K, with a 16% error. The original measurement by NA60 (AIP Conf. Proc. 1322 (2010) 1) is 205 ± 12 MeV, with a relative error of 6%, while the fit quoted in the present paper, in a different mass range, is 245 ± 17 MeV, with a relative error of 7%. This is ~ 2.5 times more precise than the averaged STAR data.

The average temperatures in the LMR and IMR from STAR data are compatible within 1.8σ . Not averaging over all data, STAR data span different baryochemical potential (μ_B) values, some of which were not measured so far. This is, of course, interesting. On the other hand, in this case, the IMR temperatures are so large that they are compatible with the LMR temperatures within 1.5σ or less.

Therefore, the main conclusions of the paper—based on a quantitatively larger temperature in the IMR compared to the LMR, signaling different sources of thermal radiation in the two regions—rest mainly on the NA60 data. It is not clear what kind of quantitative advancement we could achieve, such as improving our present knowledge of the equation of state, in a regime

of moderate μ_B , using the STAR data alone or adding them to the existing NA60 data.

The authors should explicitly acknowledge these limitations in the manuscript and discuss how they impact the interpretation of the results. They should provide a more detailed discussion of why the STAR data, despite their limitations, might still be valuable. However, they should also clearly state that the NA60 data remain the benchmark for precision and that the STAR results are complementary rather than decisive.

At the theoretical level, the microscopic many-body models, such as the one by Ralf Rapp and collaborators, developed during the 1990s, were very successful in describing the NA60 data, aligning with the main message of the present paper. NA60 data were already found to be consistent with radiation produced in the quark-gluon plasma and radiation from the hadronic phase dominated by the strongly broadened ρ meson, separated by a phase transition. The paper Nucl. Phys. A 806, 339 (2008), originally cited in the NA60 paper Eur. Phys. J. C 59 (2009) (fig. 5.1), describes at length the model and also different scenarios for the equation of state and their implications for the radiation from the partonic phase versus radiation from the hadronic phase. The more recent paper Phys. Lett. B 753 (2016) 586 makes use of a modern lattice QCD equation of state and shows that the predictions of hadronic many-body theory for a *melting* ρ meson, coupled with quark-gluon plasma emission, yield a quantitative description of the dilepton spectra at the SPS (see fig. 1).

Furthermore, as I pointed out in the previous iteration, it is an established fact since long time that the radiation from the LMR shines mostly from the medium at temperatures very close to T_c . This is reported, for instance, in J. Phys. Conf. Ser. 420 (2013) 012017 by R. Rapp, and also in Acta Phys. Polon. B 42 (2011) 2823, using the full-fledged theoretical framework published in referred journals to describe the NA60 data and cited also in the present paper.

The introduction part of the present paper provides an overview of thermal radiation produced by real and virtual photons in the hot medium produced in a high-energy nuclear collision and cites past measurements. However, it is thin in the discussion of the results in terms of their meaning from the comparison of the data to the theory along the lines discussed in the previous paragraphs (though there are references to many theoretical papers). The ρ broadening is mentioned almost in passing without a solid discussion of its

implications. To state again things in a slightly different way, the observed broadening in the NA60 data exceeds 400 MeV, approaching a divergent width—a *melting* ρ —as expected close to T_c in an almost chirally restored medium. On the other hand, this seems a novel result obtained for the first time from this analysis, while reading the section on results and discussion.

The authors should expand the discussion in the introduction and results sections to link more explicitly the findings to the existing theoretical models. References to key theoretical works, such as those by Ralf Rapp and collaborators, should be integrated more thoroughly.

Summarizing, I do not recommend the paper for publication in *Nature Communications Physics* because it does not present new results that change or improve the paradigm for the interpretation of thermal radiation produced in high-energy heavy ion collisions, which has been consolidated for the past two decades. The paper is worth publishing in more specialized journals, provided the introduction and discussion of results are revised as indicated above.

Specific comments

NA60 data are used in the present paper, and the authors remarked that it is unclear what changed from Eur. Phys. J. C59 (2009) 607 to AIP Conf. Proc. 1322 (2010) 1. I checked existing literature from NA60. First, there was an improvement in the acceptance correction procedure, which was originally multi-differential in Eur. Phys. J. C59 (2009) 607. Then, in order to improve the statistical accuracy, it was shown that it could be reduced to a p_T - M correction by integrating over other kinematic variables. I found this described in Nucl. Phys. A 783 (2007) 327.

The other aspect is the subtraction of the narrow resonances, which I commented on already in the previous iteration. An improved mass calibration allowed the narrow structures to be subtracted more precisely, leaving a smoother continuum. I found this described in Eur. Phys. J. C64 (2009) (NA60 paper on ϕ meson production). This procedure was applied to the whole LMR in AIP Conf. Proc. 1322, 1–10 (2010).